# Green gentrification in European and North American cities

Isabelle Anguelovski [1,2,3,4,16] ✉, James J. T. Connolly [1,3,5,16], Helen Cole[1,2,3], Melissa Garcia-Lamarca[1,2,3], Margarita Triguero-Mas[1,2,3], Francesc Baró [1,3,6], Nicholas Martin [1,3], David Conesa [7], Galia Shokry[1,2,3], Carmen Pérez del Pulgar [1,2,3,8], Lucia Argüelles Ramos[1,2,3,9], Austin Matheney [1,3], Elsa Gallez [1,3,6], Emilia Oscilowicz[1,3], Jésua López Máñez[10,11], Blanca Sarzo [12,13], Miguel Angel Beltrán[14] & Joaquin Martinez Minaya[15]

Although urban greening is universally recognized as an essential part of sustainable and climate-responsive cities, a growing literature on green gentrification argues that new green infrastructure, and greenspace in particular, can contribute to gentrification, thus creating social and racial inequalities in access to the benefits of greenspace and further environmental and climate injustice. In response to limited quantitative evidence documenting the temporal relationship between new greenspaces and gentrification across entire cities, let alone across various international contexts, we employ a spatially weighted Bayesian model to test the green gentrification hypothesis across 28 cities in 9 countries in North America and Europe. Here we show a strong positive and relevant relationship for at least one decade between greening in the 1990s–2000s and gentrification that occurred between 2000–2016 in 17 of the 28 cities. Our results also determine whether greening plays a "lead", "integrated", or "subsidiary" role in explaining gentrification.

[1] Institute for Environmental Science and Technology, Universitat Autònoma de Barcelona, Bellaterra (Cerdanyola del Vallès), Spain. [2] IMIM (Hospital del Mar Medical Research Institute), Barcelona, Spain. [3] Barcelona Lab for Urban Environmental Justice and Sustainability, Barcelona, Spain. [4] Institució Catalana de Recerca i Estudis Avancats, Barcelona, Spain. [5] School of Community and Regional Planning, University of British Columbia, Vancouver, BC, Canada. [6] Vrije Universiteit Brussel (VUB), Geography and Sociology Departments, Brussels, Belgium. [7] Valencia Bayesian Research Group, University of Valencia, Valencia, Spain. [8] Department Environmental Policy, Helmholtz Centre for Environmental Research (UFZ), Leipzig, Germany. [9] Universitat Oberta de Catalunya (UOC), Estudis d'Economia i Empresa and Internet Interdisciplinary Institute (IN3), Barcelona, Spain. [10] Department of Financial Economics, University of València, Valencia, Spain. [11] CaixaBank Group, Barcelona, Spain. [12] Department of Microbiology and Ecology, University of Valencia, Valencia, Spain. [13] School of Mathematics and Maxwell Institute for Mathematical Sciences, University of Edinburgh, Edinburgh, UK. [14] Fundación para el Fomento de la Investigación Sanitaria y Biomédica de la Comunitat Valenciana, Valencia, Spain. [15] Department of Applied Statistics and Operational Research, and Quality, Universitat Politècnica de València, Valencia, Spain. [16] These authors contributed equally: Isabelle Anguelovski, James J. T. Connolly. ✉email: Isabelle.Anguelovski@uab.cat

The value of urban green infrastructure and greenspace in particular is widely recognized for healthier, livable, low-carbon and climate-resilient cities. Some of the benefits include critical ecosystem services for climate adaptation, such as urban cooling and stormwater management[1–3], climate mitigation through carbon storage or sequestration of urban forests and agriculture projects[4,5], and local environmental benefits that help to manage the health effects of climate change[6–8]. Greenspaces and other forms of public places also create or reinforce community ties, civic engagement, and sense of place[9–12].

Yet, large inequalities exist and persist in the distribution of and access to greenspace, especially so by race and income, as the ample literature on urban environmental justice demonstrates. Numerous studies have identified inequities by race and class when it comes to area of accessible parks, park quality, and park maintenance and safety in the Global North, including in the United States (US)[8,13,14], France[15], Germany[16], Spain[17], and Australia[18]. Empirical research on the topic ranges from an early study of Milwaukee, which pointed at a significant positive correlation between residential tree canopy cover and median household income[19], to recent research demonstrating that place-based race, ethnicity and poverty factors are important correlates of poor spatial access to parks and other greenspaces[20–25]. These inequities in access to green infrastructure and greenspace have been linked with uneven negative ecological and climate impacts in cities[26] and attributed to a historical and social context that produced and entrenched patterns of exclusion, segregation, or unequal urban development more generally[27].

Most recently, scholars have pointed at the so-called greenspace paradox[28–30], by which seemingly laudable municipal strategies of restoring degraded urban environments, creating greenspace, or deploying climate-adaptive green infrastructure improve an area's attractiveness while resulting in increased property values, housing prices, and physical displacement of working-class residents and racialized groups and cultures—ultimately serving as a gentrifying force through a process known as green gentrification, environmental gentrification, or climate gentrification[29,31–35]. Urban greening initiatives might thus become "disruptive green landscapes"[36] or what some have called Green Locally Unwanted Land Uses for historically marginalized groups[37].

Consequently, green gentrification concerns have folded new planning conundrums into already complex environmental and climate justice challenges[31,38,39]. To the extent that interventions become embedded in processes that contribute to the displacement of the very residents urban greening was often meant to benefit, new green infrastructure can undermine municipal goals of shifting toward climate-responsive cities for all[40–43]. They can also compromise calls for transformative, equity-based climate action focusing on vulnerable groups[44,45]. In short, urban greening can become a "wicked" problem for climate and sustainability initiatives, highlighting both the presence of strong competing interests underpinned by unequal power structures[46] and the need to examine its social impacts beyond climate adaptation or mitigation benefits[47]. This quality generates the general hypothesis of the green gentrification literature: Urban greening in a neighborhood during a given time period contributes to or accelerates gentrification in that neighborhood during the period immediately following.

Thus far, all quantitative studies of green gentrification in the Global North find substantial support for the green gentrification hypothesis[41,48–58]. However, other than a few exceptions[33,48,50], these analyses do not proceed from multivariate measures of gentrification and thus do not necessarily conceptualize or operationalize the complex nexus of changes happening across social, cultural, and economic arenas within gentrification.

Rather, most quantitative studies examine single characteristics of social change as the outcome, such as changes in the percentage of wealthier residents (or residents' income), racial or ethnic characteristics, housing prices, and, in some cases, residents' occupation. Findings reveal that changes in the percentage of white residents are often revelatory indicators of green gentrification (Pearsall and Eller 2020), while other studies identify increases in housing prices[48,51,52,56–58]; in median household income[48]; in college-educated residents[48,51]; loss of African-American residents[51]; of Hispanic residents[54]; or of residents from the Global South[50] as key indicators reflecting gentrification. Meanwhile, gentrification is sometimes removed as a concept in favor of "neighborhood change" with new parks being shown to impact change variables such as residential composition, in- and out-flows of residents, housing costs, housing vacancies, and an area's reputation[53].

Beyond the conceptualization of gentrification, one important tension that remains unresolved within the green gentrification literature is the question of where and when greening plays a primary versus a subsidiary role relative to other factors fueling gentrification. Some studies expose this tension by examining the context of greening through the introduction of control variables at the (a) neighborhood level (e.g., percentage of non-white residents) and at the (b) citywide level (e.g., percentage of vacant housing). These studies, for example, point to the importance of greenspaces linked with gentrification being concentrated near existing gentrifying neighborhoods. In Philadelphia, neighborhoods with new public greenspaces located in proximity to other gentrified neighborhoods are found to be more susceptible to gentrification than those located farther away[33,49]. These studies also point to the importance of a critical density of greenspaces. In New York City, the number and percent area of greenspaces in a census tract are identified as having a strong positive impact on property values[56] and as positively correlated with gentrification[28] while this trend is not necessarily as clear with per capita income[57]. Another recent study[56] conducted in New York City included area of greenspace within census tracts as a control. Other studies, such as the US-wide study conducted by Rigolon and Németh[48] found that new parks located close to city centers tend to trigger gentrification regardless of their size and function (function is related to offering infrastructure for active transportation). Interestingly, their finding holds true for small parks, which contrasts with other studies in the US and China in which small parks are not linked with gentrification[51,55]. Last, the effect of new greenspaces has been linked with the quality of housing stock and park design. New parks in Barcelona located in post-industrial neighborhoods with rehabbed or new housing are more associated with gentrification than those located in neighborhoods with mass housing built from the 1950s–1970s[50], as are greenspaces with more esthetic and recreational value to users[59].

In this study, we address several limitations in the existing literature through our test of the green gentrification hypothesis across 28 cities in 9 countries of Western Europe and North America (Fig. 1). Our data allows for a more comprehensive analysis than previous studies on green gentrification because our cities are representative of a diversity of growth and greening trajectories; we develop a unique internationally comparative multivariate measure of neighborhood gentrification; we identify the role played by greenspaces relative to other drivers of gentrification (previously unstudied controls for other known drivers of gentrification); and we amplify the scope of analysis (citywide and across three decades). Building on this improved data foundation, we use a spatial autoregressive Bayesian model to determine the extent to which variation in new public greenspaces added during a given time period explains gentrification in the period immediately following, in the presence of covariates.

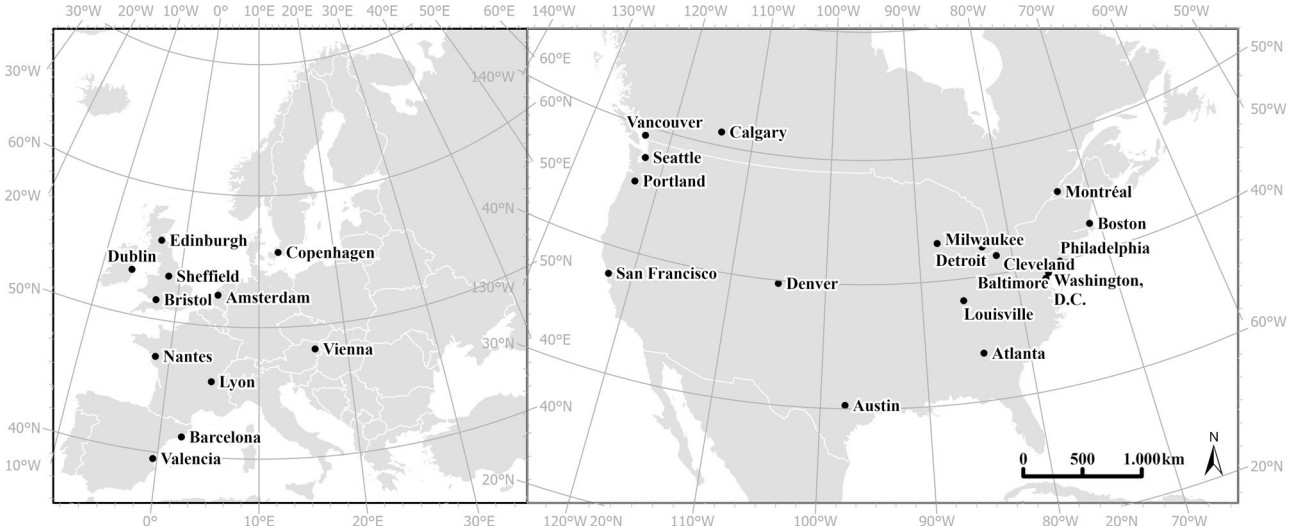

**Fig. 1 Locations of the 28 cities included in the analysis.** The cities analyzed here (labeled with points) are all mid-sized (500,000 - 1,500,000 population) and are located in 9 countries across Western Europe and North America. Cities were primarily selected to provide a diversity of geographic and growth characteristics.

We employ our model across three time periods (1990s, 2000s, and 2010s) in all neighborhoods of the 28 cities using small-area census geographies as our unit of analysis and accounting for geographic factors not specifically measured within our Bayesian models through the use of a spatial component as a conditional autoregressive term.

## Results

**Global model.** We employ a Bayesian linear mixed model (in which city and country were considered as random effects) to analyze global relationships across all 28 cities in our dataset. This global model implies that, in general, greenspaces are relevant in the presence of covariates for explaining gentrification across the entire sample during the first two decades of the 2000s ($H_2$ and $H_3$). The results, shown in Table 1, indicate a strong relevant positive relationship between greenspaces inaugurated during Periods 1 and 2 (roughly 1990–2009) and recently emerging gentrification trends during Period 3 (roughly 2010–2016). Results also indicate a positive relevant relationship across all cities considered together between greenspaces inaugurated during Period 1 (roughly the 1990s) and gentrification that unfolded across Periods 2 and 3 (roughly between 2000 and 2016). Meanwhile, the results indicate a negative, yet weakly relevant relationship between greenspaces inaugurated during Period 1 and gentrification during Period 2. This likely demonstrates that the green gentrification trend overall is a somewhat more recent phenomenon emerging with increased strength over time and some degree of lag as it unfolds, at least in a general sense when considered at the city-level across the many cities we analyze.

The other relationships indicated within the global model include a positive and relevant role for new residential development and city-level Gross Domestic Product (GDP) growth (where included) in explaining gentrification, and a negative and relevant role for closer distance from the city center. While they may have dual roles, we operationalize these variables as drivers, not consequences, of gentrification. For example, regarding GDP growth, we assume that higher economic growth may induce gentrification. It is true that it may also follow from gentrification, but we do not test this potential as we considered it outside our study framework given the hypotheses (Methods section). While these are not standardized covariates, which limits direct comparison, these results point toward a likelihood that

gentrification between 1990 and 2016 across all cities studied here was linked with new greenspace and growth initiatives in areas within the city proper and relatively close to, but not usually within, the historic city center. This greening and growth link in gentrifying areas seems to have started to be present in the cities studied here during the 2000s and into the 2010s, while prior to this time the role greening played relative to gentrification was not aligned with growth and development in a general sense.

**City-level temporal patterns of green gentrification.** While the global model offers an important overview of the overall relationships between the data in our models, a shift toward city-by-city analysis allows us to deploy a more precise Bayesian model with a spatial weight added as a conditional autoregressive term to understand the potential role of greenspace in explaining gentrification over time. Importantly, this analysis also allows us to identify areas where gentrification took place but greening was not indicated as a relevant variable in explaining it. In all, 17 out of the 28 cities show city-level results that indicate greenspace additions from an earlier time period are relevant for explaining gentrification that occurred during the period immediately following for at least one decade of the 1990s, 2000s, or 2010s. A total of 4 of the 17 cities showing green gentrification trends are European, 2 are Canadian, and 11 are in the US. In all, among the cities studied, we find that green gentrification is more prevalent in North America than Europe, but it is not a solely North American phenomenon, and tests for the effect of region and nation show that the variation is best explained at the city-level. Overall, these results that examine the effect of all new greenspaces within 28 cities show a robust occurrence of likely green gentrification outcomes for at least one decade (i.e., the majority of cities have a citywide effect indicating green gentrification occurred).

Figure 2 demonstrates two main temporal patterns of green gentrification among the 17 cities that display this trend. The first main pattern, which covers 11 of the 17 cities, indicates sustained, or long-term green gentrification across the full time period studied. In these cities, greening predicts gentrification during a period stretching over at least two decades (2000s, 2010s). These sustained temporal patterns are mostly confined to North American cities, especially cities with population and economic growth in the Sunbelt and coastal areas (i.e., Atlanta, Seattle,

**Table 1 Global model results showing strength of effect for each variable in explaining gentrification in the presence of covariates.**

| | New greenspace from prior period (same as gentrification +2) | Prior green coverage (% of area in 1990) | New residential buildings in prior period (total number) | Population change (city-level) | GDP change (city-level) | City center (distance to) | Tract density (2010) | City size (+/− 1 million) | Time from 1990 (until first time period) |
|---|---|---|---|---|---|---|---|---|---|
| H₁: Composite gentrification score, period 2 (2000s) | − | | ++ | | ++ | − − | | | |
| H₂: Composite gentrification score, periods 2–3 (2000+2010s) | + | − − | + | | | − − | | − − | |
| H₃: Composite gentrification score, period 3 (2010s) | ++ | | ++ | − − | ++ | − − | − − | ++ | − − |

The results were based on a general interpretation as follows: p > 0.50 → positive effect on gentrification; p < 0.50 → negative effect on gentrification; p - 0.50 → variable not relevant; p - 1.00 → variable is relevant and positive effect; p - 0.00 → variable is relevant and negative effect; p - 0.50 → variable not relevant; p - 1.00 → variable is relevant and positive effect. Note that new transit data is not included in global results due to a lack of information on variables across all cities. ++ strong positive effect, + positive effect, − negative effect, − − strong negative effect. Greenspace appears to be an increasing factor in gentrification over time, with relevance shifting from negative 2000s gentrification to positive for 2000–2010s gentrification and strongly positive for 2010s gentrification.

Vancouver, San Francisco, Washington, Boston, or Denver) or recovering cities (i.e., Louisville, Philadelphia, Milwaukee), with only Barcelona showing this trend in Europe. There are also six green gentrification cities (Austin, Detroit, Nantes, Edinburgh, Copenhagen, Montreal) that indicate a shorter-term temporal pattern wherein the resulting gentrification only took place over one of the three time periods, especially so in most recent years (except for Edinburgh). For these cities, green gentrification is likely present but perhaps more nascent, although limitations in data availability only allow us to draw results for Copenhagen and Edinburgh for the most recent time period, meaning we cannot say for sure in these cities whether there has been a more sustained pattern. Importantly, though, results for all of these 17 cities imply a relevant role for greening in explaining gentrification trends for at least one decade—those "short-term" cases with a more limited temporal extent of the phenomenon should not be seen as cases where there is an absence of green gentrification. For more details on the strength of the associations used to identify these patterns, see Supplementary Data.

For our remaining 11 cities, the associations we explore show negative or non-relevant relationships between greening and gentrification throughout the study period. Figure 3 shows the temporal pattern and relationships for these eleven cities (see Supplementary Data for the detailed strength of associations). Overall, those cities that present a negative or non-relevant association for citywide green gentrification are mostly European (Amsterdam, Bristol, Lyon, Sheffield, Valencia, Vienna, and Dublin). It is worth noting that in 7 of the 11 cities where greening is indicated as either negative or non-relevant for explaining gentrification (Baltimore, Bristol, Cleveland, Portland, Sheffield, Valencia, Vienna), our models include alternate variables that are potentially relevant for explaining gentrification. In these seven cities, citywide gentrification trends are linked with new development (Baltimore, Bristol, Portland, Valencia, Vienna) or new transit infrastructure instituted as part of widespread urban renewal and development (e.g., Sheffield, Cleveland), but not necessarily with greenspace per se. In other remaining cities (Calgary, Dublin in the earlier period, and Vienna in the earlier periods in particular), our models do not have a good set of relevant variables for explaining gentrification at the citywide level (although some local patterns of gentrification might exist). Last, we also note that data limitations mean we could not run an analysis for all variables/time periods for some of those 11 cities (Bristol, Lyon, Sheffield, Valencia, and Vienna in later periods).

**Greening versus other growth-oriented drivers of gentrification.** For our final result, we summarize the role played by greening relative to the other potentially relevant covariates in the city-level models. The 17 cities where we find at least one decade of citywide green gentrification can be classified differently according to the extent to which greening is indicated as a stand-alone driver of gentrification or it is enmeshed with other growth-oriented drivers, such as new residential development or new transit infrastructure. In Fig. 4, we identify three types of green gentrification cities based on the degree of relevance (i.e., stronger or weaker) and temporal extent (i.e., longer or shorter in duration) of associations implied by the city-by-city Bayesian model results. We interpret these findings as pointing toward a new typology of the relative roles played by greenspace and other variables in citywide green gentrification processes. The detailed strength and duration of associations of each of the covariates can be found in Supplementary Data.

In our first type, "Lead Green Gentrification" cities, which include 8 of the 17 identified as experiencing citywide green gentrification, greening is the standout improvement of the built

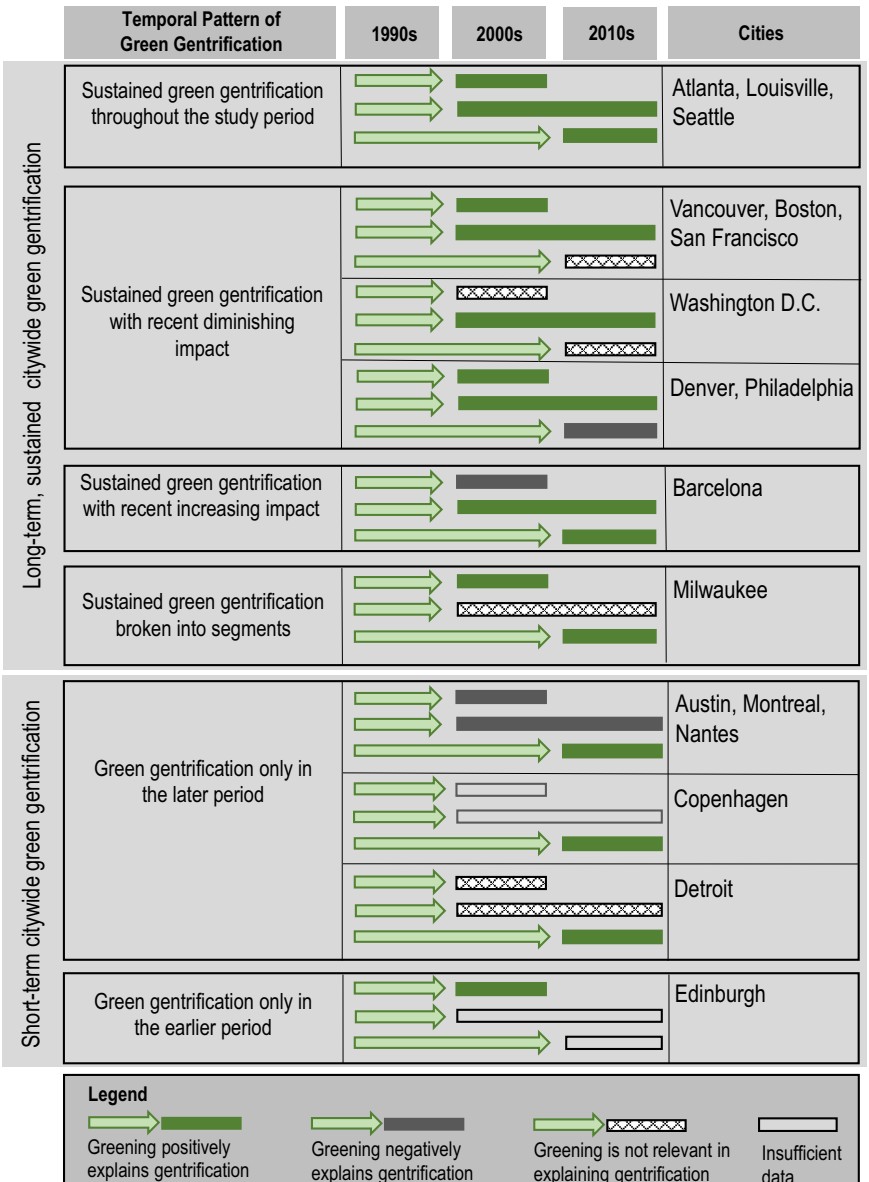

**Fig. 2 Cities with patterns of green gentrification.** These patterns show green gentrification trends over time in the 17 cities where greening from an earlier period is a likely relevant variable in positively explaining gentrification in the period(s) immediately following at some point between 1990 and 2016. There are two temporal groups: long-term (2 decades) and short-term (1 decade).

environment playing a relevant and sustained explanatory role over a long period of time in our models. While other qualities of the built environment like density and distance to center might be relevant, other changes to the built environment like new development and new transit are not, and this is the case throughout the entire study period. In short, the green gentrification hypothesis is affirmed most strongly in these cities as it appears that greening, and greening alone, is the main intervention in the built environment that explains gentrification across the city.

In our second type, "Integrated Green Gentrification" cities, which include 6 of the 17 identified as experiencing citywide green gentrification, greenspace is indicated as having an explanatory role that is roughly on par with that of new development and/or new transit in terms of the relevance and temporal extent of association. Here, green gentrification is certainly a real phenomenon, but it is not greening alone that explains gentrification. Rather, greening is mixed up within a set of local interventions

working together towards neighborhood redevelopment and, often, broader sustainability initiatives.

Finally, in our third type, "Subsidiary Green Gentrification" cities, which include 3 of the 17 identified as experiencing citywide green gentrification, greenspace is indicated as having a relevant role in explaining gentrification, yet it is a more secondary role, with other built environment interventions seemingly more prominent based on strength and duration of association. For example, in Detroit, greening is a relevant positive explanatory variable for gentrification during the 2010s, but new residential development demonstrates a more relevant and sustained explanatory role throughout the study period. In these cities, we would say that green gentrification is present, but likely plays a subsidiary or nascent role in fueling gentrification processes in the city.

Importantly, these three types of green gentrification all indicate cities where greening plays a likely role in explaining tract-level variation of gentrification results. While one might be

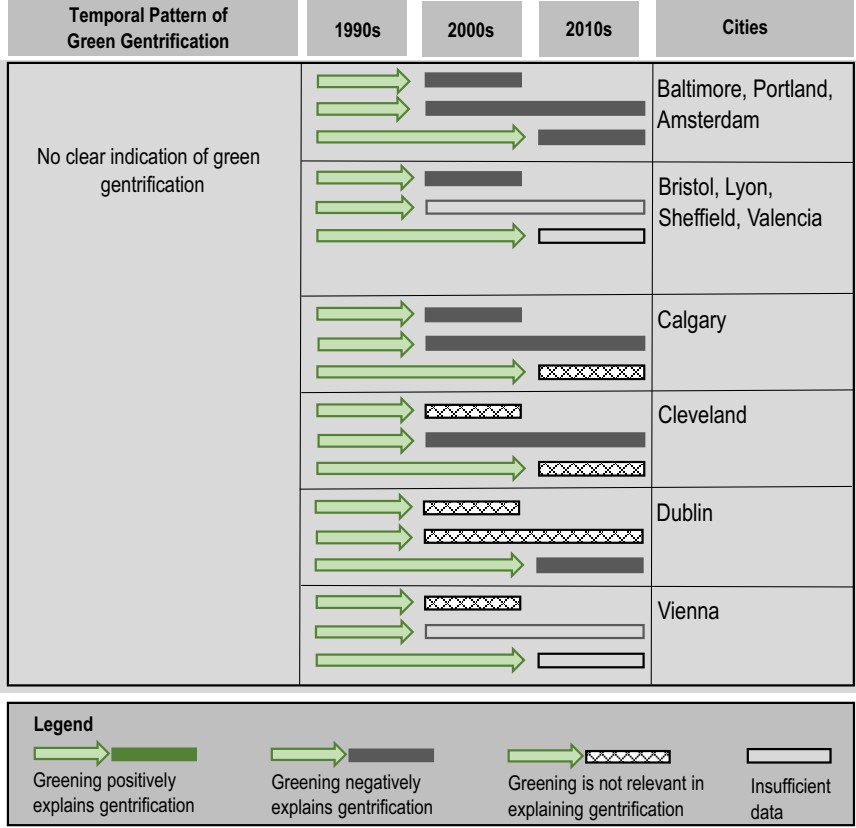

**Fig. 3 Cities with no clear patterns of green gentrification.** These patterns show cities where greening is likely either a negative or not a relevant predictor of gentrification. For these cities, new development, new transit or spatial location are other likely relevant drivers of gentrification.

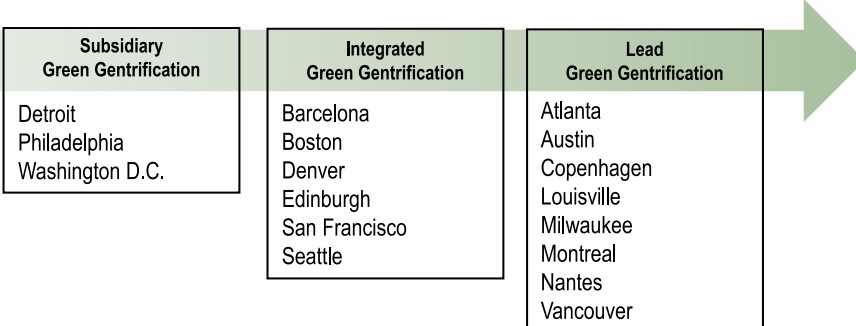

**Fig. 4 Green gentrification types. Analysis reveals three types of green gentrification cities.** In "lead green gentrification" cities, greenspace is the standout driver of gentrification. In "integrated green gentrification" cities, greenspace demonstrates is likely a relevant driver of gentrification to a degree that is roughly equal to other built environment changes, like new transit and new development. In "subsidiary green gentrification" cities, greenspace is likely a relevant driver of gentrification, but it is less impactful than other built environment changes.

tempted to discount integrated and subsidiary forms of green gentrification as less impactful, this interpretation would ignore the relevant role our model indicates for greening across all three types. In other words, it is not the case that because other potential explanatory variables, like transit and new real estate development, are also relevant that green gentrification is not occurring. Rather, it is possible that without greening as part of the mix in these cases, gentrification would not occur at all. Thus, in delineating these three types, we are highlighting the fact that greening drives gentrification in a manner that is not monolithic (and it should be emphasized that it sometimes is not associated with gentrification). However, even subsidiary green gentrification shows a likely relevant role for greening, meaning greening is implicated in the mix of forces that drive gentrification to some

degree and needs to be considered on those terms. Thus, all three forms of green gentrification are conceptually important when testing the overall hypothesis.

It is also worth noting that, while the spatial patterns of green gentrification within each city have to be considered relative to the city's local context, there are grounds for assuming that some common spatial patterns exist and that these patterns could inform future development of the typology described in Fig. 4. For example, with the exception of Nantes, distance from center always negatively explains green gentrification (usually with moderate strength) when it is relevant. This means that for the most part, we are observing processes wherein new greening, housing, and transit are sparking gentrification in areas within the city proper, and close to but generally not within the historic

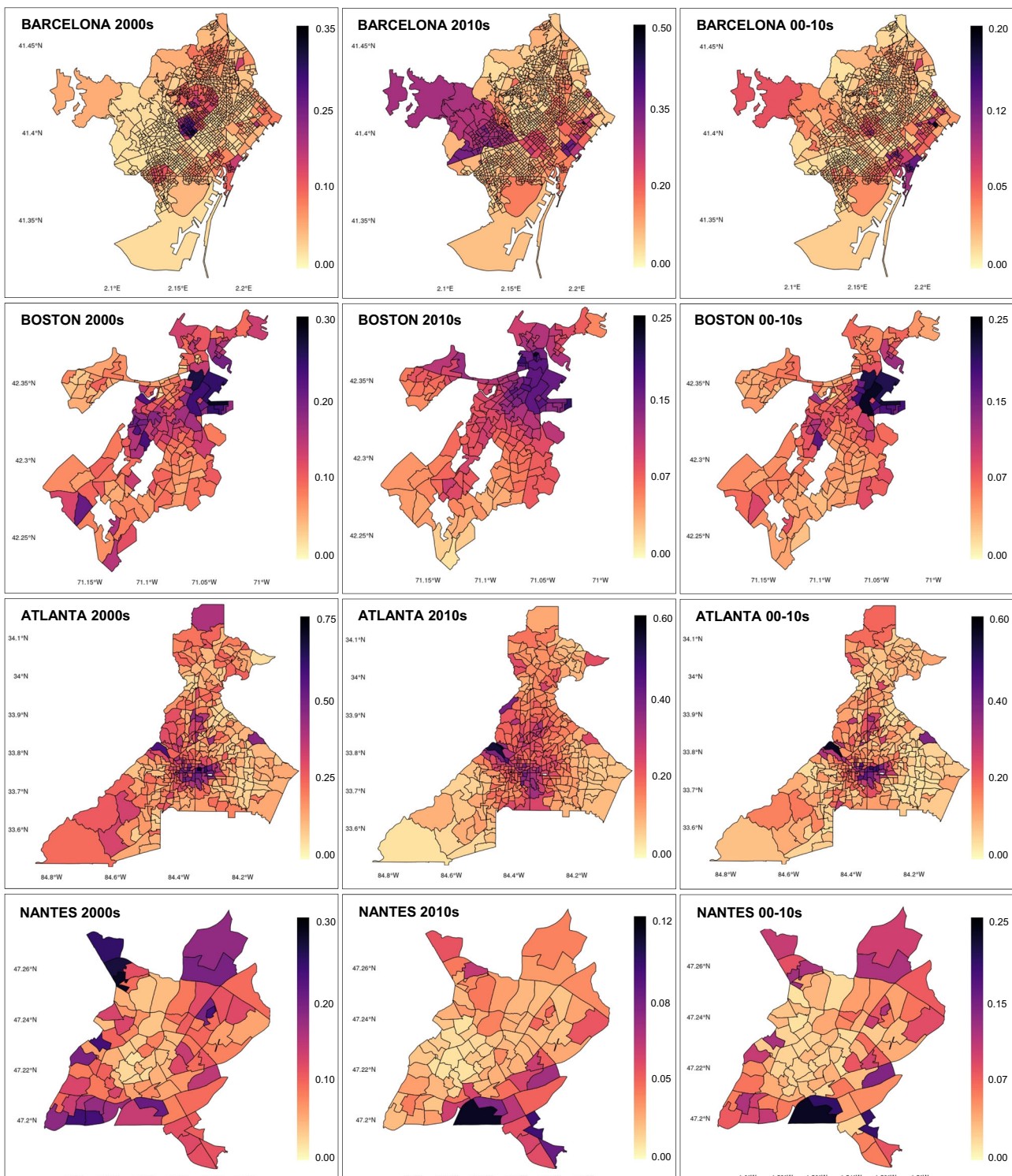

**Fig. 5 Mean of posterior predictive distributions in Barcelona, Boston, Atlanta, and Nantes.** The Bayesian posterior predictive distributions refer to the distributions of the power in predicting the gentrification index (outcome variable) in each census tract (or equivalent) produced by the final model (model that includes spatial effect and selected independent variables). They are not to be interpreted as traditional *p*-values (the probability of obtaining the observed results, assuming that the tested null hypothesis is true). Darker (darker pink, purple, and black) polygons show areas of the city where the final spatial effect model best predicts the relationship between greening and gentrification while lighter polygons (from light yellow and orange) show the areas where the final spatial effect least predicts that relationship. Because these are all green gentrification cities, high explanatory values (darker pink, purple, and black polygons) also indicate areas where green gentrification has occurred with higher likelihood. Selected cities include integrated (Barcelona, Boston) and lead (Atlanta, Nantes) green gentrification cities and include cases where the scope of green gentrification differs across time periods. Especially for lead green gentrification cities, the maps are largely indicating the specific geography of green gentrification. For integrated green gentrification cities, the maps are showing the geography of gentrification driven by a mix of factors. For example, in Nantes, for gentrification in the 2010s, the model best predicts high gentrification in the area of the Ile de Nantes (black polygon), and the surrounding areas of Vieux Malakoff, Malakoff, Champ de Mars, and Nantes Sud (darker pink and purple polygons) in the bottom center map.

downtown of the city. This often points toward post-industrial or underused areas being redeveloped adjacent to city centers. To highlight what this spatial pattern looks like in some of our cities, Fig. 5 displays the spatial pattern of the explanatory power of our model for a selected number of city types and time periods.

## Discussion

Our modeling accounting for spatial effects allows us to analyze the role of greening vis-à-vis other built environment changes in explaining citywide gentrification trends; and to assess the different role of greening across time periods. Our global model results indicate that a general association between greening and gentrification is likely a somewhat more recent overall trend in the cities under study, emerging with increased strength over time and with some lag as it unfolds, at least in a general sense when considered citywide across the cities we analyze. Moving to individual city models, in 17 out of 28 cities greening from an earlier time period is relevant for explaining gentrification for at least one period immediately following (i.e., for at least one decade) given the presence of other covariates. Those results support the overall green gentrification hypothesis within our sample given the high rate (majority of cities) and our conservative citywide analytical approach. This citywide approach offers a more robust indication than a site-level analysis examining specific greenspaces, even though those analyses offer important insights into the pathways through which green gentrification manifests[50,60]. While one might reasonably focus within our results on what we call "lead" green gentrification cities, the other types of cities wherein greening is indicated as having a less central but still relevant role in explaining the variation in gentrification trends are conceptually as important. All of these three types of green gentrification cities considered together form a heterogeneous affirmation of the overall hypothesis in many cities and point toward important nuances that need to be considered.

Overall, citywide green gentrification is mostly evident across our North American cities, but not only. In European cities with more recent green gentrification, including Copenhagen, Nantes, and Barcelona, an emerging pattern in this direction could be explained by recent context. These cities have seen rounds of investment-oriented greening and climate-responsive livability initiatives (such as the Barcelona Green and Biodiversity Plan or the Nørrebro climate resilience initiative) attract higher-income skilled workers, especially those in recently redeveloped areas. At the same time, those cities have also seen social protection and housing affordability policies—potentially protective tools against gentrification—dismantled to various degrees (e.g., Copenhagen), or such tools were almost absent until nascent efforts by more progressive mayors put housing rights for vulnerable groups at the center of urban policies, as was the case in Barcelona since 2016[61,62]. Overall, it seems green gentrification is one possible outgrowth of approaches to urban sustainability that put initiatives ostensibly focused on ecological and public health improvements in service to economic growth agendas through visibly rebranding cities as green and livable areas ripe for investment (e.g., Montreal, Austin)[61,63].

When examining the 11 cities that showed negative or non-relevant relationships between greening and gentrification throughout the study period, an important caveat is that data was not always available to examine all time periods for most of these cities. It should be noted as well that in some of these 11 cities, gentrification is shown in the existing literature to be associated with broader sustainability initiatives, potentially indicating that the citywide analytic approach used here is not capturing some important specific manifestations or that the narrower operationalization of greening as one of five types of greenspace used here may not be capturing all relevant trends (e.g., Portland)[60]. Often, these 11 cities show in our models that gentrification is mostly explained at the citywide level by new development (D) or transit (T) infrastructure. This is the case in Baltimore (D), Bristol (D), Cleveland (DT), Portland (D), Sheffield (DT), and Valencia (D). This finding resonates with events on the ground. For example, in Valencia, an epicenter of grandiose interventions during the Spanish economic bonanza of the 1990s and 2000s, gentrification has been linked to regeneration programs and the arrival of high-speed railway[64]. In other cities with negative or non-relevant relationships between greening and gentrification, other factors have been identified as strongly contributing to gentrification on a citywide scale, such as large-scale urban regeneration and high technology investment and related real estate development in places like Dublin[65]. In addition, in other cases, housing affordability and rights policies (e.g., Amsterdam, Lyon), such as 30% or more of the housing stock being allocated to social housing, might be playing an important protective role against green gentrification and its associated displacement[66]. Those anti-displacement policies are known to be more absent in North American cities, which exhibit relatively limited housing rights laws and municipal budgets heavily dependent on property taxes—and thus on increased neighborhood economic value and growth[62]. This policy context could also explain why the majority of sustained citywide green gentrification cities are located in the USA.

Looking more deeply at the types of green gentrification cities identified in Fig. 4, one series of cities (Atlanta, Austin, Copenhagen, Louisville, Milwaukee, Montreal, Nantes, and Vancouver) point toward greening, and greening alone, as the main citywide intervention relevant for explaining gentrification. These cities which we call "lead green gentrification" stand apart from others where greening plays a more integrated or subsidiary role in explaining gentrification. In Atlanta, Nantes, and Vancouver, for example, greening takes place on a large scale, throughout many areas of the city, and with active branding and green rhetoric on the part of developers and city officials[61,67] that draw new large-scale market-price housing and associated higher-income residents. Atlanta is the host of the emblematic 53 km Beltline which will eventually link 45 neighborhoods and is accompanied by new and rejuvenated parks built since the early 2000s (e.g., Old Fourth Ward, D.H. Stanton)[58,68]. Nantes with a series of green axes along its two rivers, bordering small and large parks, has since the early 2010s branded itself as "The City in a Garden"[69], building on its 2013 European Green Capital Award, with some of the most emblematic spaces located within the greened former shipyard site Ile De Nantes in the center-west area of the city, but not only.

In other cities exhibiting green gentrification—Barcelona, Boston, Denver, Edinburgh, San Francisco, and Seattle—greening plays an explanatory role roughly on par with that of new development and/or new transit. Those are "integrated green gentrification" cases. There, greening has indeed been part of an economic growth-driven redevelopment strategy, rather than a less visible, small-scale, community-oriented approach[28]. For example, Barcelona's more recent green gentrification has taken place in Sant Martí (eastern part of the city), a post-industrial area partially rebranded as the tech- and innovation-oriented 22@ district. Greening and gentrification has also taken place in the 2010s in the regenerated Old Town (Ciutat Vella)[70] and further intensified in the upper-income district of Sarrià-Sant Gervasi. In Boston, another integrated green gentrification city, several large-scale so-called resilient development projects alongside greening, which started in the mid-2000s, are located along the self-branded climate-adaptive waterfronts of East Boston, Seaport District, South Boston, and North and South Dorchester.

Much of our study's value lays in its large-scale and international scope across 28 cities in the global North, as well as our ability to assess green gentrification through a composite gentrification score, which includes both demographic and real estate change, over diverse time periods, and at a high spatial resolution. It also lays in the high analytic robustness of the study since it examines citywide, aggregate trends, allowing us to look at whether greening on the whole, as opposed to in specific sites, is relevant for explaining gentrification processes. In that sense, it is a high-bar for attaining a positive relationship between greening and gentrification and represents a conservative approach to testing the green gentrification hypothesis.

Our study also presents some limitations. First, the most recent period (2010–2016) was shortened due to data availability at the time of study, which might have affected our ability to assess the extent of gentrification in the most recent time period, thus effectively reducing its scope. While accounted for as much as possible, data availability and variables differed somewhat between the United States and Europe, which might also explain result differences. Further, some of our measures (e.g., GDP for economic growth) can be conceptualized as a consequence of gentrification, as well as a driver, which introduces some complexity to interpretation. However, the temporal structure of the model makes clear the prior position of greening before gentrification, allowing us to at least be certain that greening is wrapped up in the generative processes producing the demographic and real estate changes observed together. Finally, our study analyzes trends within formal city boundaries with the potential for modifiable areal unit problems that may come with such boundary delimitations. Future analyses must also include green gentrification trends and variations by types and size of greenspaces and by operationalizing greening in more expansive ways. Global South cities as well as smaller towns are also worthy of examination, especially so as greening is increasingly used in a variety of contexts and has become a type of global planning orthodoxy across numerous places, including cities such as Medellin or Cape Town[28,71,72].

In sum, our study shows that in most of the mid-sized cities from the Global North we evaluated, while urban greening has diverse climate, health, and socio-economic benefits, it also contributes to green gentrification processes in a number of circumstances and thus to new social, racial, and health inequalities that eventually undermine climate equity and justice. Our findings call for accompanying anti-displacement and inclusive greening policies[73], which—taken together—would ensure the construction of green and climate-responsive cities starting from an equity lens focused on long-term health equity[36,41], rather than green cities triggering or embedding dynamics of unequal urban development.

## Methods

**Overall research design and comparative framework.** This study was designed around an international, comparative analysis of mid-sized cities (500,000 to 1.5 million residents within city boundaries) in North America and Western Europe with diverse greening trajectories. Our focus on North America and Western Europe derives from a recognition of green planning traditions and trajectories that have recently influenced both regions, even if not always to the same extent[74], and an interest in comparing similar outcomes across diverse urban contexts in order to increase generalizability. The relevance of medium-sized cities is linked to the reality that such places compete for visibility in the context of transnational investment that can flow past places lacking a bold urban identity. Neighborhoods in these cities are often intense laboratories for urban regeneration, innovation, and green livability initiatives[61].

From the 99 mid-sized cities in North America and Western Europe that we initially identified, we sought to develop a final sample for analysis accounting for important differences between cities that may affect the likelihood and form of green gentrification. Our selection is not necessarily representative of all cities, but it is based on a city-by-city analysis designed to enable us to maximize geographical diversity; diversity in urban development pathways (e.g. growing or shrinking city);

and diversity across various urban forms. We prioritized this diversity approach as it provided the best means of understanding whether the dynamics we observe are simply reflective of a certain city type or region. While this does not guarantee our results apply beyond our sample, it does help to maximize generalizability. We included Northern, Continental, and Southern Europe; Western and Eastern Canada; West and East Coast US; Southern (i.e., Sunbelt) US; and Midwest (i.e., Rustbelt) US cities. In terms of urban development pathways, we included cities whose economy and population were mostly shrinking since the 1990s (e.g., Cleveland), while others economically and demographically recovered in more recent years (e.g., Philadelphia, Washington D.C.). We also selected cities with a stable, prolonged growth and economic vitality over the last 30 years (e.g., Portland; Nantes). Some cities in our final selection are compact and dense while other cities are sprawling, with important implications for the size and quantity of new greenspace as well as for the density and magnitude of real estate (re) development projects.

To finalize the selection of specific cities, we augmented these diversity criteria with a comprehensive qualitative analysis of the greening trajectory since 1990 of the 99 possible cities within our population thresholds, aiming at inclusion of many types of green trajectories so as not to only analyze cities with a strong focus on greening. We used all available public documents relevant to green plans and strategies from each city to identify the variety and diversity of green projects that were undertaken relative to neighborhood redevelopment and revitalization; and real estate changes[63]. This resulted in a classification of cities by intensity and length of their greening agenda, ranging from those seeking to be the "greenest city" or a so-called "green capital" to those for whom greening was a more muted but still highly visible goal[61]. Using this baseline of 99 cities, we then selected a final subset of 28 cities (Fig. 1) with the most even representation across all of our diversity criteria balanced against the extent of available data at the census tract level (or equivalent) on green project creation, demographic change, and real estate values from 1990 to the mid-2010s (Table 2).

Our goal in completing an extensive three-decade long (1990s, 2000, 2010s) analysis stemmed from the fact that the 1990s are widely considered as the starting point for formal urban sustainability and later climate planning programs in many cities around the world before their explosion in the 2000s and 2010s[75–77]. Initiatives such as the 1987 Brundtland Report by the United Nations, the 1992 Rio Declaration on Environment and Development, its complementary Agenda 21 (action plans intended for implementation at various scales), and the 1997 Habitat Agenda rapidly catalyzed the start of sustainability and climate planning in cities.

The broad temporal, geographic, and urban typological scope highlights the intent of our study to indicate the general degree to which green gentrification is occurring in the Global North. This intent justified the use of a conservative city-level measure wherein we only report green gentrification when the cumulative effect of all greenspaces across a city has a relevant role overall in explaining gentrification trends, using the census tract as the unit of analysis. Requiring that it be reflected citywide is a high-bar for testing the green gentrification hypothesis because there could easily have been instances of gentrifying tracts around one or a few greenspaces without registering a city-level trend if there were enough sites where green gentrification did not occur in the rest of the city. We deemed it important to maintain this conservative, citywide approach because we sought a fundamental test of the overall green gentrification hypothesis. An alternate approach focused on individual sites would almost certainly expose specific instances of green gentrification within a greater number of cities, but would not necessarily test the overall effect of new greenspaces.

**Data selection and collection.** In order to build the models for this analysis, for each city we gathered data on greenspaces, gentrification, and covariates (i.e., control) variables. The data on greenspaces includes the spatial boundaries for all parks, greenways, preserves, gardens (formally sanctioned or not), or recreation areas in place as of 2016, and the year of inauguration of the spaces—starting in 1990. Acknowledging that there are multiple definitions and understandings of greenspace across disciplines (Taylor and Hochuli, 2017), for the purpose of this study we define it as the creation of any of our five classifications of urban greenspace—parks, greenways, greenways, preserves, gardens, or recreation areas (for detailed description see ref. [24]). In some cases, the boundaries and sizes of greenspaces represent a manually augmented version of publicly available files when missing spaces were identified through other sources. To identify the year of inauguration, we used a combination of direct communications with city or non-profit organization staff, city land records, published reports, media reports, historic imagery, and city archival searches. We defined inauguration as the year the space was acquired for public use as a greenspace. If this year was not available, then the year the space was opened to the public was used instead.

The gentrification data includes changes in demographics and real estate values reported at the highest spatial resolution possible. These data are derived from national and local statistics offices (see Supplementary Table 1 for a full list of sources and data cleaning approaches). In all cases, the data were reported at the smallest area unit used by that country or city for all variables (e.g., census tract in the United States, IRIS in France, and Secció Censal in Barcelona). For each city, we searched for the most closely aligned demographic variables that corresponded to constructs commonly used to measure the socio-cultural dimensions of gentrification, in addition to rent changes (see Supplementary Note and

**Table 2 Basic descriptive characteristics of cities included in the analysis.**

| City | Population | Population change | | | % GDP growth (yearly average growth) | |
|---|---|---|---|---|---|---|
| | 2016 | 1990–2000 | 2000–2010 | 2010–2016 | 2001–2010 | 2011–2016 |
| Amsterdam | 821,800[a] | + | + | + | 1.5 | 1.7 |
| Atlanta | 479,200 | + | + | + | 0.6 | 3.7 |
| Austin | 939,400 | + | + | + | 3.5 | 4.5 |
| Baltimore | 616,200 | − | − | − | n.a | n.a |
| Barcelona | 1,608,700 | − | + | + | 1.5 | 0.2 |
| Boston | 679,800 | + | + | + | 1.5 | 2.6 |
| Bristol | 567,100 | + | + | + | 1.7 | 1.6 |
| Calgary | 1,239,200 | + | + | + | n.a | 4.2 |
| Cleveland | 387,700 | − | − | − | n.a | n.a |
| Copenhagen | 591,500 | + | + | + | 1.1 | 2.3 |
| Denver | 696,200 | + | + | + | 0.7 | 3.7 |
| Detroit | 677,100 | + | − | − | −1.4 | 3.0 |
| Dublin | 554,500 | + | + | + | 3.2 | 4.0 |
| Edinburgh | 512,500 | + | + | + | 2.2 | 1.6 |
| Louisville | 617,600 | − | + | + | n.a | n.a |
| Lyon | 513,300 | + | + | + | 1.6 | 1.6 |
| Milwaukee | 597,000 | − | − | + | 0.7 | 1.2 |
| Montreal | 1,942,000 | + | + | + | n.a | 1.9 |
| Nantes | 446,500 | + | + | + | 1.3 | 2.6 |
| Philadelphia | 1,576,000 | − | + | + | 1.6 | 1.7 |
| Portland | 642,700 | + | + | + | 1.7 | 2.9 |
| San Francisco | 871,500 | + | + | + | 1.2 | 5.7 |
| Seattle | 709,600 | + | + | + | 2.4 | 4.3 |
| Sheffield | 541,800 | - | + | + | 1.5 | 1.6 |
| Valencia | 790,200 | − | + | + | 1.7 | −0.3 |
| Vancouver | 631,500 | + | + | + | n.a | 3.0 |
| Vienna | 1,856,600 | + | + | + | 1.4 | 0.0 |
| Washington DC | 685,800 | − | + | + | 3.1 | 1.3 |

Sources: United States Census Bureau, Statistics Canada, Organization for Economic Co-operation and Development (OECD), Eurostat.
[a]2015 Population.

Supplementary References for a full description of the literature used to develop the variable list). These included relative demographic changes demonstrated by the (1) change in socially vulnerable population; and relative change in social class as defined by: (2) population with a high level of education, (3) population with relatively high incomes, (4) population with a professional occupation, and (5) population classified as living below poverty level. To operationalize our determination of socially vulnerable, we developed a custom measure of socially vulnerable population variables such as race, ethnicity and migration status. Data on socially vulnerable populations varied relative to the local context. For example, in some cities scholarly research supports race and ethnicity data as a key indicator of social vulnerability, but in others measures of immigration or migration are the key marker of social vulnerability. In response, we developed bespoke measures of socially vulnerable populations by city relative to available data and existing scholarly findings. In Valencia, for example, we included percentage of residents with nationality from countries in Africa, Philippines, Peru, Pakistan, Bolivia, Ecuador, Colombia, or the Dominican Republic (see next section for the general description of gentrification variables and Supplementary Methods, Supplementary Table 2 in particular, for exact variables used in each city). In most cases, the small-area geographies used to report the data changed across years. In order to standardize the boundaries for analysis across time, we used a hierarchically ordered set of techniques designed to minimize error as much as possible (see description of the "CUDA" approach to spatial data standardization in Supplementary Methods, section on Data Processing in particular, for a complete description of the process used to standardize data).

The covariate data includes small-area and city-level controls. At the small-area level, we gathered data on the location of all new transit stops added since 1990, the distance from city center, the residential density, and the number of new residential buildings constructed for each time period analyzed. In order to identify the number of new residential buildings constructed in each small-area geography by decade, the number of new buildings was derived from national or state statistics office reports, local building permit datasets, and lot-level tax or cadastral files with "year built" designations in each lot (see Supplementary Table 3). For the transit stops, we used Google maps, Wikipedia and city-level planning documents to identify the location of all new rail and bus stops added since 1990. For distance to the city center and population density, we made these calculations manually from the centroid of each small area using ArcGIS 10.6 desktop software. At the city-

level, we gathered data on population change since 1990, city-level GDP change since 2001 (due to data availability), and prior green coverage (in 1990) expressed as percent of city area. For city-level population change since 1990 and city-level GDP change, we used World Population Review and OECD data respectively. For prior green coverage, we calculated this value manually using ArcGIS 10.6 desktop software.

**Data processing for greenspace and gentrification measures**. In order to calculate the amount of greenspace added over time to a given small-area geography, it was first assumed that every greenspace has a catchment area wherein nearby residents are easily drawn to it. We employed the widely used measure of 400 m as an easily walkable (https://www.healthyactivebydesign.com.au/design-features/public-open-spaces/evidence/#distancequal) (~5–10 min) estimate of a standard catchment for a greenspace. For this reason, we calculated the number and area of greenspaces in a given small-area geography by first creating a 400-m Euclidean distance buffer around the area. We then used the formal tract boundary plus the 400-m buffer for all greenspace calculations by tract (see ref. Connolly and Anguelovski (2021)[24,62] for a more detailed discussion of this procedure).

Next, we calculated the area of greenspace added during each of three time periods. Owing to data limitations, these periods approximate but do not always exactly align with decadal splits. For example, Time Period 1 in a given city may cover 1992–2001 while it covers 1990–2000 in another city. In order to account for this variation, we customized the allocation of greenspaces to time periods according to the gentrification data coverage. Total area of greenspace added during a time period was calculated as the area of all greenspaces inaugurated during the period covered by the gentrification data plus that added in the two years immediately following that period. The 2-year overlap was added to account for "announcement effects" where spaces announced in the time immediately prior (but presumably built within 2 years after the start of the period) are also included[78]. For example, if the gentrification data for Time Period 2 in a given city measures changes between 1999–2006, then the Period 2 greenspace data would be the sum of area added between 1999 and 2008.

In order to test our main hypothesis—greenspace in a prior period explains gentrification in the period immediately following—we developed a gentrification score for each of three time periods based on a diversity-weighted sum of demographic change plus change in rent values. As change in rent is an indicator of

market dynamics and prices, rather than of ability of the population to pay, change in rent and change in income are distinct measures. To develop the score, we used data from each city that coincides with a generalized list of five potential social change indicators chosen to reflect variables of importance that capture some of the complexity in the gentrification literature (for detailed variables used in each city and discussion of the literature supporting that variable, see Supplementary Note and Supplementary Methods, Supplementary Table 2 in particular). These indicators are:

*nvul*—percent of residents from a non-vulnerable ethnic group
*uni*—percent of residents with a university degree or higher
*prof*—percent of residents classified as having a professional occupation
*hises*—percent of residents classified as having a high income
*npov*—percent of residents not classified as being below the poverty level
Additionally, a single indicator for including the influence of rent values was also incorporated:

*hrent*—percent of households paying above median value in rent
Next, we took several steps to combine these measures into a composite gentrification score standardized in a way that makes them comparable across national and city borders. This composite score (Eq. 1) is defined as:

$$G_{\text{tract}} = H_{\text{E}}(Z_{\text{nvul}} + Z_{\text{uni}} + Z_{\text{pro}} + Z_{\text{inc}} + Z_{\text{pov}}) + Z_{\text{rent}} \qquad (1)$$

where $G_{\text{tract}}$ is the composite gentrification score for a given small-area census geography; $H_{\text{E}}$ is Shannon's diversity index standardized to a score between 0 and 1 as the weight applied to the social change variables; $Z_{\text{x}}$ is the standardized score of the given social change variable; and $Z_{\text{rent}}$ is the standardized score for the change in rent variable.

Our formula calculates $Z_{\text{x}}$ as the change in the value over each time period normalized by the citywide change for that same time period. This normalized measure controls for the fact that we would expect relatively high small-area changes in cities with consistently high rates of change and the opposite in cities with consistently low rates of change. In order to make these normalized measures of change for each variable comparable across cities, we calculated their $Z$-values as the number of standard deviations that an observation is above or below the mean value of observations for the city. The result was a measure of magnitude of change across five socio-cultural (including race, ethnicity and immigration status in *nvul*) variables and one rent variable suitable for comparison across cities.

Having generated the foundation of our gentrification score, we next created a weight for the diversity of change occurring. This weight is meant to respect the complexity of gentrification processes as expressed in the literature wherein we consider it to be most acute and most accurately identified in neighborhoods where there are multiple social, economic, and cultural changes happening at once[80,81]. Thus, in our view—and as other studies of green gentrification posited[50]—a greater diversity of change across multiple variables is more indicative of gentrification than changes within only one or two variables. We added a weight for this type of diversity in change by calculating Shannon's Equitability Index ($H_{\text{E}}$) for the five social variables—a unique theoretical and methodological contribution of our study in the analysis of gentrification.

Shannon's Equitability Index ($H_{\text{E}}$) is a variation of Shannon's Index developed in prior studies and typically used to provide a measure of both the abundance and the evenness of a group of observations in species diversity studies[79]. The Shannon Index is calculated by summing the product of the proportional increase in each indicator in relation to the increases in all social change indicators at that location and the negative natural logarithm of that increase for each indicator variable at each location across the city. The "equitability" variation on the original index allows the number of variables measured per indicator to vary, which was necessary for us as not all cities had all five social variables available. In our case, we used this approach to develop a weight that gives greater importance to small-area units that have change across many variables at once, rather than only having change across one or two, in order to better represent the complexity and diversity of gentrification as a process of urban change[80,81].

Finally, with the standardized and normalized version of the variables weighted by the Shannon's Equitability Index, we calculated a final gentrification score. For our analysis, the extent to which gentrification is occurring is measured as the sum of the standardized and normalized changes in social variables weighted for the diversity of change occurring plus the standardized and normalized change in our rent indicator (*hrent*). A higher value from this equation indicates that more demographic and real estate changes are occurring at once in the area, which we use as a proxy measurement for the degree of gentrification. Thus, we use here a unique scale of gentrification activity in small-area geographies that can be compared across national and municipal contexts.

**Quantitative models.** Our aim in the study was to examine the relationship between greening and gentrification globally and then city-by-city, through testing our main hypothesis across different time periods. Given the temporal structure of our data, this resulted in three sub-hypotheses:

**H₁:** New greenspaces attributed to time period 1 are relevant in the presence of covariates for explaining gentrification that occurred during time period 2.

**H₂:** New greenspaces attributed to time period 1 are relevant in the presence of covariates for explaining gentrification that occurred during time periods 2 and 3.

**H₃:** New greenspaces attributed to time periods 1 or 2 (compounded) are relevant in the presence of covariates for explaining gentrification that occurred during time period 3.

With the final aim of incorporating the underlying variability between the different cities and countries, a linear mixed model (in which city and country were considered as random effects) was used for the global analysis. The model also included all covariates previously mentioned.

As performing a complete model selection for linear mixed models can be computationally challenging, we used a two-step procedure to identify the (best) final model. First, after removing those variables with substantial missing data, a Bayesian model selection procedure for multiple linear regression models with fixed effects (*BayesVarSel*[82]) was used to show which covariates were the ones that better explained the response (in presence of the others). Owing to the importance of some variables (such as greenspace), this selection process started from the null model that incorporates them.

Second, starting from the selected model for each hypothesis, we selected the best model among those with the covariates already included, plus the two random effects, *country* and *city*. Country was found to be not relevant in all the analyses. This non-relevant role for country shows that all the underlying variability was due to differences between cities (a relevant effect in all hypotheses).

Inference on the final global model of each hypothesis was performed within the Bayesian framework and so prior elicitation for all the parameters and hyperparameters of the model was required. Inference is performed within the Bayesian statistical approach as it allows us to deal in a natural way with the uncertainty of both the parameters and the models. In particular, we chose vague prior distributions to emphasize our lack of knowledge about them. The complete global model can be expressed as follows (Eq. (2)), where $n = 28$ cities:

$$y_i \sim N(\mu_i, \sigma_e), i = 1, \ldots, n$$
$$\mu_i = V_i\beta + s_i$$
$$\beta_1, \beta_2, \ldots, \beta_{p+1} \sim N(0, \sigma^2 = 1000) \qquad (2)$$
$$s_i \sim N(0, \sigma_s^2)$$
$$\sigma_e, \sigma_s \sim pc - \text{priori}(2, 0.5)$$

With respect to the city-level analyses for each hypothesis, a three-step procedure was used to check which variables of interest were relevant and whether there was any spatial effect (showing possible hidden geographic drivers resulting from a wide range of possible dynamics within the underlying urbanization process). Again, as a first step, *BayesVarSel*[82] was used to show which covariates were the relevant ones (in presence of the others). Next, we used the Moran index[83] to test whether or not the residuals of the model presented spatial autocorrelation. When they did, our third step was to add a spatial random effect that aimed to explain the underlying spatial effect. The spatial component was included as a conditional autoregressive term, reinforcing the idea that the value of an area depends on the values of its neighbors. In order to avoid confounding of the spatial effect and the covariates, the model was expressed as a spatially restricted regression where the spatial random effect is orthogonal to the covariates space.

As before, inference was performed within the Bayesian framework and prior elicitation for all the parameters and hyperparameters of the model was again required. In particular, we chose vague prior distributions to emphasize our lack of knowledge about them. The complete model can be expressed as follows (Eq. (3)):

$$y_i \sim N(\mu_i, \sigma_e), i = 1, \ldots, n$$
$$\mu_i = V_i\beta + s_i$$
$$\beta_1, \beta_2, \ldots, \beta_{p+1} \sim N(0, \sigma^2 = 1000)$$
$$s_{i|j} \sim N\left(\frac{1}{k_i}\sum_{i \sim j} s_j, \frac{\sigma_s^2}{k_i}\right), i \neq j \qquad (3)$$
$$\sigma_e, \sigma_s \sim pc - \text{priori}(2, 0.5)$$

As the posterior distributions of the parameters and hyperparameters do not have an analytical expression, all the resulting models were fitted using the Integrated Nested Laplace Approximation methodology, which is a computationally fast software implementation that provides accurate approximations to posterior distributions[84]. An important note is that Bayesian probabilities are not traditional *p*-values and should not be considered as interchangeable.

**Reporting summary**. Further information on research design is available in the Nature Research Reporting Summary linked to this article.

## Data availability

All non-proprietary data generated during the study and analyzed within the study will be available in public repository being developed by the Barcelona Lab for Urban Environmental Justice and Sustainability and scheduled for release in early 2024. This repository will be online and publicly accessible. Until the online repository is fully developed, the national statistics data used is available from the various countries' national statistics offices. The greenspace and other processed data is available upon reasonable request to the corresponding author, Isabelle Anguelovski. The raw

Copenhagen and Vienna demographic and development data are protected and are not available due to data privacy laws. The US demographic data was also purchased. The raw data itself is available free of charge, but the version we used was pre-processed to a standardized geography by a company, GeoLytics, into the 2010 census tracts on which our demographic analysis was based.

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

## Acknowledgements
Horizon 2020 (European Research Council) GreenLULUS (GA678034). Spanish Ministerio de Ciencia e Innovación—Maria de Maeztu (CEX2019-000940-M). Spanish Ministerio de Ciencia e Innovación—PID2019-106341GB-I00 (jointly financed by the European Regional Development Fund, FEDER). Spanish Ministerio de Ciencia e Innovación—Juan de la Cierva Incorporación (IJC2020-046064-I). Spanish Ministry of Economy and Competitiveness—Juan de la Cierva Incorporación program (IJC-2018-035322-I). Banco Santander-UAB Talent Fellowship program.

## Author contributions
I.A. and J.C. co-led the study, jointly supervised, co-conceived and designed the study, co-led analysis, interpretation and writing; They are both joint first authors. D.C., J.L.M., B.S., M.A.B., J.M.M. performed Bayesian calculations; H.C., M.G.L., M.T.M., F.B., N.M., G.S., C.P.P., L.A.R., A.M., E.G., E.O. developed the data, analyzed, discussed the results, wrote the paper, and contributed to the revision of the final manuscript.

## Competing interests
The authors declare no competing interests.
