## [Peer Review File · Nature Communications]

Reviewer comments, first round -

Reviewer #1 (Remarks to the Author):

The research paper responds to an emergent and significant urban issue, with a focus on the potential negative effects at a city-level, of increased green urban infrastructure, for social and racial equity.

The paper presents original quantitative analysis of urban greening and gentrification across 28 cities in western Europe and North America. While the country contexts are limited (ie they do not include primary data for other continents) the theoretical and methodological contributions of the paper are considerable – as are the findings in relation to the countries included in the study.

In short, the paper is timely, original, robust in methods and scope, and provides a quality account of a significant research and urban greening planning matter, that has global significance. The paper is high quality in its presentation and content throughout.

The data and methods used in the analysis are impressive in scope and depth, use of standard measures of walkable green space and management of data from multiple sources in ways to enable detailed modelling of 30 years of spatial data, across contexts.

The presentation of results including summary figures and maps is clear and succinct throughout, and highly accessible for readers.

A major conceptual and methodological advance developed in the research paper is identification of different types of nature-based green gentrification, distinguishing between 'lead' gentrification and other subsequent types of green gentrification processes.

As per the scope of the paper, the focus is on the Global North and in mid-sized cities.

My only comment for author consideration in what is otherwise an extremely robust, wide-ranging and well-executed and communicated piece of notable original research, is that reflection for future directions warrants reflection, in the conclusion of the paper. This would include, for example, reflection on the relevance of the findings for urban greening in Global South contexts, as well as for smaller and larger city locations.

Overall, this research paper is impressive in its contribution, method, timeliness and original contribution and I recommend it for publication with minor changes as per previous comment on reflecting on the wider significance of the research findings.

Reviewer #2 (Remarks to the Author):

The paper, "Urban greening and gentrification: quantitative evidence from 28 Global North cities," was overall a strong paper. Some noteworthy findings include evidence of greening-induced gentrification, creation of a typology of cities experiencing greening and gentrification, and nuanced findings of effects of greening relative to other processes on gentrification. This work is original and will be of significance to the field, as it expands on prior work and provides others with methods for classifying cities as well as calculating gentrification scores. The work supports the conclusions and claims being made by the authors.

There were no explicit flaws in the analysis, though I have some clarifying questions/requests:
- Some detail is needed regarding the winnowing process from 99 to 28 cities. This process is only described vaguely, but was an important part of the study, as it determined which cities were investigated. Please either describe it further here or in supplementary material (depending on word limits)

- Please provide clarification regarding "special variables relative to the local context" (lines 218-219). Just a few examples following "e.g." would be helpful.
- How did you define the five gentrification indicators? Quick definitions would be helpful.
- The way that you modified the Shannon Index is good, but it is unclear why this modification necessitates the term 'equitability'. What about this modification has to do with equitability (just curious and you may want to explain)?
- Regarding hypothesis 3 (lines 339-40), are you assessing the compounding effects of greening from both time periods 1 and 2? Or trying to isolate effects of greening that happened in either time period on Period 3?
- Please provide some justification for excluding from your equations the covariates that do not explain the response as well (line 350)
- Just a disclosure that I have very little experience with Bayesian statistics and thus cannot comment on the model expressions or other components of Bayesian analysis. However, I do think that, given the wide readership of Nature Communications, you may want to display your models in a figure form, in addition to these equations.
- It is hard to differentiate what is a cause of gentrification and what is an indicator. Lines 403-405: What makes these explanatory instead of a consequence of gentrification? Please either discuss in the limitations the difficulties in separating causes from consequences of gentrification, or justify why these are not results of gentrification.
- Figure 6 is a bit hard to read. What is in the parentheses following the city names? Is this a caption of the three panels per city? And the hue scale is hard to see when it is so little. You could consider reducing the number of cities to allow each city to be larger and easier to see the effects.

Lastly, in the future, the authors are encouraged to increase the amount that they cite literature outside of their own lab, as it reduced the anonymity of this paper.

Overall, a good paper. The authors should be commended for their efforts.

-LEM

Reviewer #3 (Remarks to the Author):

General comment

Many thanks for the opportunity to review 'Urban greening and gentrification: quantitative evidence from 28 Global North cities'. The aim of this paper is to address methodological (causal) shortcomings in some of the literature in order to understand whether urban greening in period t_n explains change in a composite gentrification index in t_{n+1} and/or t_{n+2} . A panel of 28 global north cities is set up.

Overall, the aim of the paper is a valuable area of research and one where clarity is needed. Greener cities are required for a number of the reasons laid out in the paper, at the same time there is a concern that unless managed appropriately the urban green agenda may reinforce or re-articulate existing inequalities – they author(s) 'wicked problem statement'.

Methodologically, I also think that the authors are on the right track (but also see comments below) when attempting, I'm simplifying, to say causality must mean one thing happens first, then the next thing happens. To this effect the authors have set up, what appears to, a reasonable data set.

Major comments

However, I am not sure I feel that the authors have achieved their aim or adequately discussed their results in light of the question that they ask. Does greening in one period lead to gentrification in subsequent periods? The results, as reported in the paper, are not clear. The authors argue that in 17 of the 28 cities greening precedes gentrification and therefore urban greening strategies need to be carefully managed. I do have great sympathy with the argument, but I do not feel that the evidence presented is sufficiently strong to say anything about greening

per se. First there are 11 cities where there is no or a negative effect. Rounding a bit the evidence is this almost 50:50. Second, the authors then find that in only 8 of the 17 cities (that is just under a third of the sample) 'greening is the standout environmental improvement playing a relevant and sustained role' (p.23). In the remaining green gentrifying cities greening is on par or a secondary explanation. If taking a cautious approach and adding these not-the-main-culprit cities to the 11 negative cities then typically urban greening is not leading gentrification.

The authors do provide a number of explanations for why the 11 cities may not show the expected green gentrification outcomes, but this seems somewhat ad hoc. One could presumably, construct similar ad hoc arguments for why the 8 cities where greening precedes gentrification in reality reflects something else. This analysis – and the categorisation of cities – would, in my view, also be considerably strengthened by providing additional information on decision making. How is strength measured of relationships measured? I find the explanation for the p values in the supplementary material quite confusing. Only once is it greater than 1. The explanation is that this measures the probability of a coefficient being greater than 0. Should this be different from 0 (in which case it is interpreted as a conventional p-value...which would change the conclusions significantly)? Further detail on how the strength of associations is measured/determined would help readers assess the evidence.

This something else brings me to another concern. The authors find that urban greening leading to gentrification is more typical of north American cities than European cities. Again, this suggests that the evidence itself does not lend itself well to generalising about causality and effect of urban greening. The authors rightly point out that there are important contextual differences between north American cities and European, and I agree. The problem is that these differences are not really controlled for in the analysis and so difficult to assess in terms of explanatory validity.

As presented the conclusions and results seem to better fit the first half (emphasis added) of the following conclusion on p24 than the second part. "It is also worth noting that, while the spatial patterns of green gentrification within each city appear to be unique and related to the city's local context, there may also be grounds for assuming that more universal spatial patterns exist that could inform future development of the typology described in Figure 5". The evidence as presented is not strongly indicating universal spatial patterns. I would thus like to see a more critical reflection of the results against the aims and the RQ of the paper.

A third and final major concern is a lack of any discussion of the very large literature on gentrification and urban dynamics change across urban studies, housing studies and urban and regional economics. In general, the research design around gentrification seems driven by data availability/opportunity, rather than theory. For a study focusing on causal effects, this is a major omission. For instance the gentrification index includes demographic change, but demographic base lines (prime earnings age) are also associated with income changes/human capital changes so that 10 years later gentrification would be measured without any actual change of neighbourhood composition. See for instance Rosenthal 2008 on neighbourhood cycles.

Minor comments

p.1 line 26 – is this intended as the title of the paper? If so, the title here differs from the title above the abstract.

p.7 line 165 – on what basis was the sample reduced from 99 to 28 cities?

p.8 Table part – should this be GDP (yearly average growth)?

p.9 line 202 – I could not find the details on the 5 classifications of urban space.

p.12 line 281 – is not per cent of households paying above median rent a measure of income – i.e. doubles up for hises?

p.16 line 374 – there is little later discussion of autoregressive variables in the results section. Was this not found relevant?

p. 18 line 405 – the interpretation of the distance to CBD variable is confusing. If the dependent variables is the gentrification index and there is a negative coefficient on distance to the CBD, then the interpretation is that inner cities gentrify more than outer cities. If I've understood the variable constructions correctly, then discussion of the distance (i.e. statements like "gentrification between 1990 and 2016 across all cities studied here was linked with greening and growth initiatives in areas within the city proper but outside of the historic city center") need to be revised. Moreover, this raises questions about drivers of neighbourhood change more generally.

p.18 Figure 2 – are the hypothesis is reported differently here than earlier in the paper?

Title: Urban greening and gentrification: quantitative evidence from 28 Global North cities

2 February 2022

Dear Reviewers,

Thank you very much for the opportunity to revise our paper “Urban greening and gentrification: quantitative evidence from 28 Global North cities” to Nature Communications. Thank you for also your close attention to our paper and for your recommendations for further improvement. The specific responses we made in response to your comments and recommendations points are discussed below. We have highlighted our main edits in yellow throughout our paper.

Response to Reviewer 1

Thank you very much for carefully reading the manuscript and suggesting areas for improvement. We have addressed all your points in the following way (We follow here the order of your suggestions):

Reviewer Comments	Author Responses
In short, the paper is timely, original, robust in methods and scope, and provides a quality account of a significant research and urban greening planning matter, that has global significance. The paper is high quality in its presentation and content throughout.	Many thanks for your overall evaluation of the paper.
My only comment for author consideration in what is otherwise an extremely robust, wide-ranging and well-executed and communicated piece of notable original research, is that reflection for future directions warrants reflection, in the conclusion of the paper. This would include, for example, reflection on the relevance of the findings for urban greening in Global South contexts, as well as for smaller and larger city locations.	We have added a reflection about the importance of research in those geographies/contexts in the final section of the paper as well as relevant references (line 663-666)

Response to Reviewer 2

Thank you very much for carefully reading the manuscript and suggesting areas for improvement. We have addressed all your points in the following way (We follow here the order of your suggestions):

Reviewer Comments	Author Responses
- Some detail is needed regarding the winnowing process from 99 to 28 cities. This process is only described vaguely, but was an important part of the study, as it determined which cities were investigated. Please either describe it further here or in supplementary material (depending on word limits)	Thank you for this comment. The process of selecting cities for study is an important consideration with direct impact on our findings and your comment has led us to further refine how we describe this process. As we explained (lines 143-145), we primarily sought to capture a diversity of cities. We attempted to balance diversity of regions/geographies, of urban development pathways, of urban forms, and of green trajectories against data

	availability for green data and demographic and real estate data. In this revision, we further expand on this explanation to clarify key criteria. (lines 150-178).
- Please provide clarification regarding "special variables relative to the local context" (lines 218-219). Just a few examples following "e.g." would be helpful.	We specified what we meant by special variables (lines 227-244). For example, measures of immigration or migration as marker of social vulnerability had to be used in lieu of race/ethnicity in cities in which race/ethnicity was not measured in the data. In Valencia, for example, we included percentage of residents with nationality from countries in Africa, Philippines, Peru, Pakistan, Bolivia, Ecuador, Colombia, or the Dominican Republic – see next section for the general description of gentrification variables and the supplementary material for exact variables used in each city).
- How did you define the five gentrification indicators? Quick definitions would be helpful.	We first prepared a protocol of indicators and their definition, based on existing literature and past published measures of gentrification (see the new Section 1 of the Supplemental material describing existing conceptualization and operationalization of gentrification). This became the standard for gathering and preparing the census and other administrative data from each city/country. The process of selecting variables and standardizing them as relative measures is described in Section 2 of the Supplemental Materials, which we have also updated. Briefly, using the protocol as a guide, the study team then searched for the most appropriate and available data at the smallest geographic area scale for each city. We then identified the available variables that best matched these indicators and calculated, for the small geographic areas, a relative measure (such as percent per small area). In addition to the detailed information found in the Supplemental Material, we now include a brief description of these 5 indicators in the text where we first discuss the gentrification data (lines 231-244).
- The way that you modified the Shannon Index is good, but it is unclear why this modification necessitates the term 'equitability'. What about this modification has to do with equitability (just curious and you may want to explain)?	The "equitability" term and adaptation is not ours (e.g., see Magurran, 'Ecological diversity and its measurement' (1988) (http://dx.doi.org/10.1007/978-94-015-7358-0). Rather, it is a further method devised for using the Shannon Index concept when comparing datasets of different numbers of observation categories (as in the present study). Normalizing the Shannon Index values is achieved by dividing them by the natural log of the number of observation categories/indicators. This approach represents one of several available ways of measuring evenness across different ecosystems that contain varying numbers of species. In this case, it represents the evenness of changes among a series of observed social indicators and was deemed to be the most appropriate approach for our purposes. Finally, it is important to note that the use of the word 'equitability' here in no way refers to issues of social equitability. It merely refers to the concept of evenness or 'sameness' among the changes observed. We have further clarified these aspects in lines 332-334.
- Regarding hypothesis 3 (lines 339-40), are you assessing the compounding effects of greening from	It is the compounding effects from both 1 and 2. We added a brief clarification in the text (see line 367)

both time periods 1 and 2? Or trying to isolate effects of greening that happened in either time period on Period 3?	
- Please provide some justification for excluding from your equations the covariates that do not explain the response as well (line 350)	This is an interesting point raised. It is always important to have a well-justified criteria for choosing among differing explanations (i.e., covariates) for a response variable. In this paper, as we explained in Section 4o the methods, we propose the use of a Bayesian model selection approach that chooses the simplest model (in line with Occam's razor) that best fits the data. It is based on the Bayes factor (Kass and Raftery, 1995) that reports the evidence in the data favoring each of the models and can be easily translated to posterior probabilities. It chooses among all the possible models that can explain the response by selecting the one with higher posterior probability. As a result, more than excluding covariates, we select the model that best explains the response variable of interest.
- Just a disclosure that I have very little experience with Bayesian statistics and thus cannot comment on the model expressions or other components of Bayesian analysis. However, I do think that, given the wide readership of Nature Communications, you may want to display your models in a figure form, in addition to these equations.	Thanks for your comment, but we have decided to not add a figure since we were already at a maximum number and the ones we have included are essential for the comprehension of the study.
- It is hard to differentiate what is a cause of gentrification and what is an indicator. Lines 403-405: What makes these explanatory instead of a consequence of gentrification? Please either discuss in the limitations the difficulties in separating causes from consequences of gentrification, or justify why these are not results of gentrification.	-This is an interesting comment and part of broader discussions in the gentrification literature. In line 433-436, we have now clarified that in this study and in models, we conceptualized and used these variables as drivers, not consequences of gentrification. For example, we are using these measures as proxies for whether higher economic growth (as GDP) before and during gentrification periods explains the gentrification results. We have added a brief discussion about this point in the Limitations section of the Discussion and recognize that it is a fuzzy aspect in gentrification studies (659-661)
- Figure 6 is a bit hard to read. What is in the parentheses following the city names? Is this a caption of the three panels per city? And the hue scale is hard to see when it is so little. You could consider reducing the number of cities to allow each city to be larger and easier to see the effects.	We understand this point and have decided to simplify Figure 6 for the reader by 1) focusing only on a few lead and integrated gentrification cities 2) leaving out the scale and exclusively displaying the color scheme/intensity of the relationship 3) clarifying the significance of the maps in the caption. See new Figure 6 and its legend.
Lastly, in the future, the authors are encouraged to increase the amount that they cite literature outside of their own lab, as it reduced the anonymity of this paper.	This a fair comment. We may have leaned a bit too heavily on our prior research because this has been a strong developing storyline for us, so our work here relates closely to our prior work, but your point is well-taken. We have added additional, external references from gentrification studies in the literature review, in the Discussion, and in our Supplementary Materials (see yellow highlights of references). We appreciate you raising it.

Response to Reviewer 3

Thank you very much for carefully reading the manuscript and suggesting areas for improvement. We have addressed all your points in the following way (We follow here the order of your suggestions):

Reviewer Comments	Author Responses
Does greening in one period lead to gentrification in subsequent periods? The results, as reported in the paper, are not clear. The authors argue that in 17 of the 28 cities greening precedes gentrification and therefore urban greening strategies need to be carefully managed. I do have great sympathy with the argument, but I do not feel that the evidence presented is sufficiently strong to say anything about greening per se. First there are 11 cities where there is no or a negative effect. Rounding a bit the evidence is this almost 50:50. Second, the authors then find that in only 8 of the 17 cities (that is just under a third of the sample) ‘greening is the standout environmental improvement playing a relevant and sustained role’ (p.23). In the remaining green gentrifying cities greening is on par or a secondary explanation. If taking a cautious approach and adding these not-the-main-culprit cities to the 11 negative cities then typically urban greening is not leading gentrification. The authors do provide a number of explanations for why the 11 cities may not show the expected green gentrification outcomes, but this seems somewhat ad hoc. One could presumably, construct similar ad hoc arguments for why the 8 cities where greening precedes gentrification in reality reflects something else. This analysis – and the categorisation of cities – would, in my view, also be considerably strengthened by providing additional information on decision making.	We understand your interpretation that because a bit less than half of cities do not show strong GG, the GG conclusion should be questioned. However, because our results are based on aggregate city-wide level data, the relationship between greening and gentrification has been examined in a highly rigorous and conservative manner, with an overall high rate of green gentrification given this aggregate approach. For those other cities, there might actually be instances of green gentrification at the neighborhood level (and other papers written by us and by other researchers have uncovered those instances), but this city-wide aggregate approach sets up a much higher bar to uncover GG. Thus, given that our analysis is conducted at the city-wide level, which we see as a strength of our study, we encounter a high prevalence of GG and in a diversity of cities. In short, it is much more difficult to find a positive relationship at the city-scale size than at the neighborhood level. Last, we also note that insufficient data limited our ability to run analysis for some of those 11 cities (Bristol, Lyon, Sheffield, Valencia, and Vienna in later periods), meaning that aggregate-level GG might be taking place, but that we do not have enough data to run this analysis. We edited our text accordingly on lines 461-465 and in the Discussion (563-565 + 571-579; 650-654). Last, what you call “ad hoc” explanations about what might be happening in those other cities are contextual interpretations also derived from field work and other analyses conducted in those cities. We attempted to provide a few more details/examples (and references) in some of those cities (607-609).
How is strength of relationships measured? I find the explanation for the p values in the supplementary material quite confusing. Only once is it greater than 1. The explanation is that this measures the probability of a coefficient being greater than 0. Should this be different from 0 (in which case it is interpreted as a conventional p-value...which would change the conclusions significantly)? Further detail on how the strength of associations is measured/determined would help readers assess the evidence	As you highlight, it is very important to measure the possible relationships between our variable of interest and the covariates and spatial effects. In our case, as mentioned in the paper, we have chosen a three-step procedure to determine which covariates influence gentrification and whether there was a spatial effect in each city. From the first step, we have used a Bayesian model selection approach that chooses the simplest model (in line with Occam’s razor) that best fits the data. This involves choosing among all the possible models that can explain the response by selecting the one with the highest posterior probability. As a result, we provide a list of the covariates of interest and then we check the spatial effect. In the Tables of the Supplementary Material (section 4.2), we present the results of those covariates and the spatial effect, and

the way they affect the response variables of interest. But we have also included a measure of the strength of those covariates with the response. In our case, as we are working with a Bayesian approach, we have decided to measure them through means of the posterior probability that the parameter (the coefficient of the covariate) is positive.

Posterior probability of being greater than 0

As it can be seen in this simple figure, the posterior probability of being greater than 1 is large (close to 1). This indicates that the relevance of that covariate is great, that is, it has a strong relationship.

It is worth noting that all probabilities should be between 0 and 1. No values greater than 1 should appear. We have decided to clarify any possible misunderstanding, and so we have slightly modified values of $2e-04$ that indicates 0.0002. We have also changed one table with typos on it, as there were negative probabilities (which are clearly not possible).

Finally, we want to highlight that these probabilities are quite different from traditional p-values, and should not be interchanged. As above mentioned, the value of p corresponds to the probability that the parameter is greater than 0 and so, large values imply relevant positive effects while small values imply relevant negative effects.

For length reasons, we have decided not to include these comments in the current form of the paper, although we do clarify that Bayesian probabilities are not traditional p-values and should not be interpreted as such (line 414).

The authors find that urban greening leading to gentrification is more typical of north American cities than European cities. Again, this suggests that the evidence itself does not lend itself well to generalising about causality and effect of urban greening. The authors rightly point out that there are important contextual differences between north American cities and European, and I agree. The problem is that these differences are not really controlled for in the analysis

As stated by the reviewer, the differences between American and European cities are not controlled for in the analysis. Our analysis controlled for the cities (showing that there are differences between them), and for countries (showing no differences between them). We have clarified this accordingly along the new version of the paper (383-385; 462-464).

and so difficult to assess in terms of explanatory validity.	
As presented the conclusions and results seem to better fit the first half (emphasis added) of the following conclusion on p24 than the second part. "It is also worth noting that, while the spatial patterns of green gentrification within each city appear to be unique and related to the city's local context, there may also be grounds for assuming that more universal spatial patterns exist that could inform future development of the typology described in Figure 5". The evidence as presented is not strongly indicating universal spatial patterns. I would thus like to see a more critical reflection of the results against the aims and the RQ of the paper.	As mentioned above, we are looking at city-level aggregated greening, so our results push toward a high level GG effect conclusion. Yet, we also agree that it is important to highlight that GG is not happening everywhere. Rather than universal, the word "common" is probably better positioned describe the trends. This trend happens often, but not always. We modified our wording on line 546.
A third and final major concern is a lack of any discussion of the very large literature on gentrification and urban dynamics change across urban studies, housing studies and urban and regional economics. In general, the research design around gentrification seems driven by data availability/opportunity, rather than theory. For a study focusing on causal effects, this is a major omission. For instance the gentrification index includes demographic change, but demographic base lines (prime earnings age) are also associated with income changes/human capital changes so that 10 years later gentrification would be measured without any actual change of neighbourhood composition. See for instance Rosenthal 2008 on neighbourhood cycles.	Thank you for raising this issue as it was one we struggled with in drafting the text. The space limitations of the article made inclusion of the extensive literature review we completed impossible within the text itself. We struggled to decide how much to try to include. In the end, we agree that we excluded too much. In response, we have added some additional clarifications in the text (only minor) and also added into the Supplementary Materials a full new brief literature review explaining the primary literature upon which we relied to conceptualize gentrification and build a baseline list of quantitative variables with which we could measure it. This new lit review in the supplement is referenced in the text itself as well, so readers are aware that the deeper dive into the literature is available.
p.1 line 26 – is this intended as the title of the paper? If so, the title here differs from the title above the abstract.	We have clarified that this is the title of the Introductory section. We have also edited the title. Please see the new version of the paper.
p.7 line 165 – on what basis was the sample reduced from 99 to 28 cities?	Thank you for this comment. The process of selecting cities for study is an important consideration with direct impact on our findings. As we explained (lines 143-145), we sought to capture a diversity of cities. We attempted to balance diversity of regions/geographies, of urban development pathways, of urban forms, and of green trajectories against data availability for green data and demographic and real estate data. In this revision, we further expand on this explanation to clarify key criteria. (line 153-178).
p.8 Table part – should this be GDP (yearly average growth)?	Yes, this refers to yearly average growth. Please see the last column of the Table (we have clarified this point).
p.9 line 202 – I could not find the details on the 5 classifications of urban space.	This classification is presented in the second sentence of the subsection called "2) data selection and collection"
p.12 line 281 – is not per cent of households paying above median rent a measure of income – i.e. doubles up for hises?	We included both those paying above median rent and a measure of high socioeconomic status as we see these as distinctly different. The cost of rent is not based on the income of those living in an area, but rather on a price/value driven by the housing market. In

	fact it is the difference between ability to pay (based on income) and the cost of rent that ultimately leads to displacement in many cases. We have added a brief description of this distinction in the text in lines 293-294 + 305-308.
p.16 line 374 – there is little later discussion of autoregressive variables in the results section. Was this not found relevant?	The importance of the spatial effects and its presence in the final models was quite relevant. Indeed, many of the cities have a spatial effect (based on an autoregressive model). Comments about this can be found at subsection 4 of the Methods, in particular lines 398-400.
If the dependent variables is the gentrification index and there is a negative coefficient on distance to the Central Business District, then the interpretation is that inner cities gentrify more than outer cities. If I've understood the variable constructions correctly, then discussion of the distance (i.e. statements like “gentrification between 1990 and 2016 across all cities studied here was linked with greening and growth initiatives in areas within the city proper but outside of the historic city center”) need to be revised. Moreover, this raises questions about drivers of neighbourhood change more generally.	Thank you for pointing out the unclear way in which we discussed this finding. We have revised the language used throughout to make it clear that the negative coefficient indicates closer to historic center is predictive of higher gentrification scores. The nuance we sought to maintain is that it usually indicates areas close to, but not inside of historic centers (439-440).
p.18 Figure 2 – are the hypothesis is reported differently here than earlier in the paper?	We made a mistake in transferring text from our data files and did not immediately notice. We apologize. We adjusted Figure 2 and reversed the numbering of the hypotheses.

Reviewer comments, second round -

Reviewer #2 (Remarks to the Author):

Thank you for addressing reviewer concerns so thoroughly. I appreciate the depth of attention paid to feedback, and am pleased with the result of revision. As far as I am concerned, this paper is good for publication.

Reviewer #3 (Remarks to the Author):

Many thanks for the opportunity to re-review 'Urban greening and gentrification ...' and for the additional edits/material made by the author(s). While the additional material on gentrification (supplementary material) does begin to bring in some of the literature on urban dynamics processes I am afraid I remain unconvinced by the paper's central ability to do what it sets out to do – establish causality in the change processes/ gentrification.

There is a large literature that looks at urban spatial change processes (e.g. Rosenthal 2008, Brueckner and Rosenthal 2009); environmental factors and urban change (Kahn and Walsh 2014), agglomeration and urban dynamics (Ahlfeldt and Pietrostefani 2019), and natural amenities and neighbourhood dynamics (Brueckner et al 1999, Lee and Lin 2017) and little of this is reflected in the model set-ups or discussion of results. For the global models the variables found consistently significant are new house building and distance to central place. One could thus also construe an argument where greening initially did not lead to gentrification (period 1), but economic and structural changes nevertheless resulted in population re-centralisation/inner city revival. Once in swing (non-industrial) employees demanded local amenities reflecting their lifestyles and amplifying property values (weak in period 2, stronger in period 3). Precisely because of the importance of causality in this area I think that the research as a whole is valuable and important. I am just not convinced by how the evidence is interpreted or motivated.

I also remain concerned about the process of getting from 99 to 28 cities. I take the author(s) point that they have aimed for diversity and geographic spread, but coming from a different quantitative background that the one use in the paper I am wondering what this selection process does to the relative weight that individual cities exert in the analysis. The sample does not appear to be aiming for representativeness of North American and European cities, as such it seems ambitious to use the results to extrapolate/infer meaning beyond the sample itself.

Similarly, I remain unconvinced by the author(s) reply to the 8/17 out of 28 comment in my earlier feedback. I fully accept that evidencing gentrification at a city-wide scale is a challenge (also raises questions about what is actually captured if the city as a whole changes; and what the functional boundaries of cities are), but unlike the authors I feel that this raises the burden of proof, rather than lessening it. If in 8 out of 28 cities, in the author(s)' view, there is strong evidence for environmental improvement driving gentrification then this, to me, is not a common trend in the selected sample.

Finally, the models in the supplementary material differ in variable selection. Again, different methods traditions deal with these things differently, but in light of the large literature on urban change/processes it seems peculiar that in some cases only a very small number of variables are significant determinants of change; it is also not clear how the endogeneity between green space early 2000s and change processes in subsequent periods is dealt with – or put differently, are the authors confident that the green space variables are orthogonal to everything else that is omitted from this models? Urban systems change slowly. The fact that a variable might be measured marginally before other variables (including change variables) is this not in and of itself sufficient for the orthogonality assumption to hold.

I appreciate that a lot of work has gone into collecting and preparing data and many of my observations appear to arise from thinking differently about how causality is evidenced/explored in

an urban context. In my reading I remain unconvinced that the paper achieves what it sets out to do, but others may feel differently.

Title: Urban greening and gentrification: quantitative evidence from 28 Global North cities

19 April 2022

Dear Editor,

Thank you very much for the opportunity to revise our paper “Urban greening and gentrification: quantitative evidence from 28 Global North cities” to Nature Communications. The specific responses we made in response to the comments and recommendations are discussed below. We have also highlighted our main edits in pink throughout the main body of our paper.

Response to Reviewer 3

Thank you very much for carefully reading the manuscript and suggesting areas for improvement. We have addressed all your points in the following way (We follow here the order of your suggestions):

Reviewer Comments	Author Responses
Similarly, I remain unconvinced by the author(s) reply to the 8/17 out of 28 comment in my earlier feedback. I fully accept that evidencing gentrification at a city-wide scale is a challenge (also raises questions about what is actually captured if the city as a whole changes; and what the functional boundaries of cities are), but unlike the authors I feel that this raises the burden of proof, rather than lessening it. If in 8 out of 28 cities, in the author(s)' view, there is strong evidence for environmental improvement driving gentrification then this, to me, is not a common trend in the selected sample.	While we see that the interpretation that is offered here would lead to the conclusion that is suggested, we believe that we may have framed our results in a manner that has led to an overly narrow interpretation of the findings. We have tried to clarify our justification for our interpretation in our revised text and figures. Specifically, lines 555-567; and lines 600-605 reflect our revisions. We also changed the wording we use to describe the temporal differences in order to better reflect how we understand these findings (see paragraph beginning on line 475). Overall, we reaffirm but further justify our interpretation that greening is a relevant driver of citywide gentrification processes using census tract as the unit of analysis in 17 out of 28 cities. We clarify that the reason to differentiate the 8 cities we identify as “Lead” and that are highlighted by the reviewer is that they show a more robust pattern of relevance for greening than the other 9 green gentrification cities. But, this does not bely the fact that the results are relevant for all 17 cities and that, in fact, the conceptual importance of all three categories is equivalent when addressing the overall green gentrification hypothesis. As we write in the revision: “While one might be tempted to discount integrated and subsidiary forms of green gentrification as less impactful, this interpretation would ignore the relevant role greening plays in these types. It is not the case that because other variables, like transit and new development, are also relevant that green gentrification is not occurring. Rather, it is possible that without greening as part of the mix in these cases, gentrification would not have occurred. Thus, in delineating these three types, we are highlighting the fact that greening predicts gentrification in a manner that is not monolithic (and it should be emphasized

	that it sometimes does not predict it at all). But, even subsidiary green gentrification shows a relevant role for greening, meaning greening is implicated in the mix of forces that drive gentrification to some relevant degree and needs to be considered on those terms. Thus, all three forms of green gentrification are of equivalent conceptual importance when testing the overall hypothesis.” As a further clarification in response to the “raises questions” comment, when we speak of the “citywide” dimension of the results, what we mean is that, in order for a city to be reported by us as showing green gentrification trends, it has to show more than just one or two instances of green gentrification within specific tracts (the unit of analysis) – rather, it has to be the dominant trend with regard for greening across the city. This goes much further than most green gentrification studies focused on smaller geographies, including one or a few neighborhoods or a few local parks at a time. In short, it demands that, for the period reported, greening considered in the presence of other explanatory factors is, on the whole, a primary explanatory variable when all tract-level instances of gentrification within the whole city are considered. That this occurs for at least one decade within 17 cities is, we argue, an important finding of a majority trend. The issue of city boundaries that the reviewer raises is certainly one that the field of geography has dealt with for decades and the potentiality of modifiable areal unit problems is partially muted by the inclusion of a spatial autoregressive term. This is, though, a reasonable limitation to call out, and we now report this issue as a limitation (but not one that invalidates any results).
There is a large literature that looks at urban spatial change processes (e.g. Rosenthal 2008, Brueckner and Rosenthal 2009); environmental factors and urban change (Kahn and Walsh 2014), agglomeration and urban dynamics (Ahlfeldt and Pietrostefani 2019), and natural amenities and neighbourhood dynamics (Brueckner et al 1999, Lee and Lin 2017) and little of this is reflected in the model set-ups or discussion of results. For the global models the variables found consistently significant are new house building and distance to central place. One could thus also construe an argument where greening initially did not lead to gentrification (period 1), but economic and structural changes nevertheless resulted in population re-centralisation/inner city revival. Once in swing (non-industrial) employees demanded local amenities reflecting their lifestyles and amplifying property values (weak in period 2, stronger in period 3). Precisely because of the importance of causality in this area I think that the research as a whole is valuable and important. I am just not	Thank you for pointing at these other studies and traditions to understand urban spatial change. We reviewed them and would like to point out important nuances between these studies and ours – After this review we have chosen to include some of the suggested studies in the supplementary material. 1) The dependent variable of the cited studies often seems to be location of higher-income households rather than a more complex understanding of gentrification per se (as a composite variable), which we consider is a limit of these studies and which also reflects the tradition in economics (the field of all these studies) to tend to look at income as marker of urban socio-spatial change. Importantly, we reflect a wider tradition of gentrification studies, which highlights that income may be a reasonable proxy for the complicated socio-cultural dynamics embedded in processes of gentrification, but sometimes you may have processes wherein income really does not shift (at least for a while) but other cultural markers like race, ethnicity, education do shift. In sum, while we see the value of the studies cited, we draw from a different, and what we believe, more nuanced and complex conceptual base in developing our gentrification measure.

convinced by how the evidence is interpreted or motivated.

More specifically, in the urban planning and geography literature from which we draw – and these are the fields in which our study is anchored – gentrification is understood as a complex and multifaceted process which is best measured through the use of a composite or index variable made of several indicators of gentrification, as we point out at in the Supplementary Materials. Because, as the reviewer acknowledges, this is along, large, and well-established debate with many existing reviews, we do not mention this full literature in the core body of the paper.

Perhaps most importantly, in a rather short text like that for a journal like Nature Communications, discussing the theoretical ramifications between the literature on urban spatial change and gentrification is beyond the scope (and not necessary for our core variables to be conceptually supported). In this study, we had to be quite targeted in presenting what results and variables show while packing a lot into the results we report (by building a gentrification index, for example). However, in our supplementary materials, we do briefly mention the newer studies the reviewer mentioned and acknowledge their contribution (page 2 and 3) and the different traditions that are present.

2) We also want to highlight the point that, as we see it, some of the studies listed by the reviewer (Lee and Lin, Ahlfeldt and Pietrostefani) are not studies of gentrification per se. Those studies examine other dimensions of urban spatial change (density elasticity of rents, density elasticity of construction cost) as outcome variable. Knowing the research question, outcome variable, and theoretical contribution of our paper, we look at gentrification as a whole, not at specific types of residents for example. And our analysis remains a city-wide analysis where we draw conclusions on specific cities where the addition of green space in period X is relevant for explaining gentrification in following period(s), while taking confounders into consideration.

3) Last, we see where the reviewer is going and get the point with the interpretation of the possible explanation of our model result, but would highlight an important missing element. Overall, the important finding is that greening is, for the 17 cities we test, wrapped up in the processes that generate gentrification. Certainly, there can be a dynamic wherein gentrification leads to greening (a perhaps oversimplification of the point being made by the reviewer), but that would be a separate hypothesis and separate model. What we test and find is that greening, at least in part, leads to gentrification based on the temporal structure of our model. Does gentrification then lead to more demand for greening? Probably yes in many cases, but that is a separate question. Overall, the explanation of urban change as presented is possible, BUT we are not able to draw such conclusions with the data, HOs, and models we have. In this study, we are looking at city-wide dynamics, NOT specific dynamics between neighborhoods and inner-city spatial

	urban processes. Our research question and quantitative data raise these questions, which we agree need further analysis in other studies, but do not point to the finding suggested. In the Discussion, we try to point toward the dynamics described here and do hypothesize what might be happening in some cities, but this is at the level of possible further directions mobilizing findings of other published studies, as we make clear in the Discussion. It is by no means a quantitative result from our core analysis here. We have tried to clarify this in the text, specifically see lines 686-690.
I also remain concerned about the process of getting from 99 to 28 cities. I take the author(s) point that they have aimed for diversity and geographic spread, but coming from a different quantitative background that the one use in the paper I am wondering what this selection process does to the relative weight that individual cities exert in the analysis. The sample does not appear to be aiming for representativeness of North American and European cities, as such it seems ambitious to use the results to extrapolate/infer meaning beyond the sample itself.	We understand your concern. However, as we explain, we do not aim at a statistical representativeness of a broader population of cities, as we now specifically clarify (see new text on line 156-163). Our choice of cities is driven both by theoretical and data availability (at the smallest geographical unit) concerns. We acknowledge in the text that this approach does not necessarily mean that results apply outside of our sample, but it does help to maximize generalizability. We have also reviewed our paper to ensure that we are not extrapolating results beyond our sample of 28 cities (see for example lines 699-701 “our study shows that in most of the mid-sized cities from the Global North we evaluated”). Put differently, we always add a disclaimer or a specification about the fact that our results are valid for our sample of cities (see for example lines 592-593, 608) and not for an entire population of cities. In general, because of the theoretical variation we introduce in the cities, we want to highlight that this does provide some element of generalizability without providing any definitive evidence beyond our sample. We do so in the context of a finding of green gentrification in 17 out of 28 cities within our sample, which we argue is a robust result, especially knowing the conservative, city-wide approach we took.
Finally, the models in the supplementary material differ in variable selection. Again, different methods traditions deal with these things differently, but in light of the large literature on urban change/processes it seems peculiar that in some cases only a very small number of variables are significant determinants of change; it is also not clear how the endogeneity between green space early 2000s and change processes in subsequent periods is dealt with – or put differently, are the authors confident that the green space variables are orthogonal to everything else that is omitted from this models? Urban systems change slowly. The fact that a variable might be measured marginally before other variables (including change variables) is this	This is a very interesting point raised by the reviewer. In order to avoid the usual confounding problem when a spatial effect is introduced in the model, we decided to add the spatial effect for each model in an orthogonal way (see for instance Martínez-Beneito and Botella-Rocamora 2021, for a detailed description of the confounding problem in areal data). The way to do so, as commented in the paper, is a two-step process:  1. We firstly selected the variables that best explain the process. We do realize other traditions might rather choose to “force” all variables to stay in the model, but our understanding (as statisticians) is that choosing final variables based on statistical fit rather than theoretical assumptions is a better way to do so. This sometimes provide final models with a very small number of variables, that we think that these are the

not in and of itself sufficient for the orthogonality assumption to hold

ones that better explain the behavior of those particular cities for that HO.

2. In second place, we added a spatial effect orthogonal to the space of the covariates to explain spatial residuals. Thus, in the cases where other relevant covariates were available, we were able to incorporate them before adding the residual spatial effect to the model.

But, most importantly, our model still has an error component (which does not have a spatial structure) which encompasses the residuals that could comprise other possible effects (of covariates) not included. As a result, other possible green-space variables do not need to be orthogonal. Orthogonality appears for each model as a helpful tool to avoid the possible spatial structures already expressed by the selected covariates.

Reference:

M. A. Martínez-Beneito and P. Botella-Rocamora (2021). Disease Mapping. Taylor and Francis, Boca Raton.

Reviewer comments, third round -

Reviewer #2 (Remarks to the Author):

Dear Authors,

Thank you for your efforts to respond to reviewer concerns. I appreciate the thorough response both in the paper and in the memo to reviewers. Well done.

Reviewer #3 (Remarks to the Author):

Many thanks for the opportunity to re-review 'Urban greening and gentrification: quantitative evidence from 28 Global North cities' and many thanks to the authors for taking the time to respond with great diligence to my earlier sets of comments.

While I remain unconvinced about aspects of this – in part because in my training the methodology for dealing with hypothesis testing differs substantially – I can agree with the author(s) that "greening is implicated in the mix of forces that drive gentrification to some relevant degree and needs to be considered in those terms" is a defensible conclusion based on the analysis and evidence produced.

I also acknowledge numerous additional qualifications around generalisability of the findings / conclusions.

My main issue is with respect to what goes before what and how this is dealt with (methodologically), including sample selection, but I think this remains an issue now for the editors to consider. The responses from the author(s) are considered and reflective and, while ultimately not leading to much change, do lead to additional qualifications. I can also accept that different fields draw on different sets of literature, conceptualizations and ultimately methods.

Title: Green gentrification in European and North American Cities
(slightly modified new version according to NCOMMS title structure requirements)

30 May 2022

Response to Reviewer 3

Reviewer Comments	Author Responses
Many thanks for the opportunity to re-review 'Urban greening and gentrification: quantitative evidence from 28 Global North cities' and many thanks to the authors for taking the time to respond with great diligence to my earlier sets of comments. While I remain unconvinced about aspects of this – in part because in my training the methodology for dealing with hypothesis testing differs substantially – I can agree with the author(s) that “greening is implicated in the mix of forces that drive gentrification to some relevant degree and needs to be considered in those terms” is a defensible conclusion based on the analysis and evidence produced. I also acknowledge numerous additional qualifications around generalisability of the findings / conclusions. My main issue is with respect to what goes before what and how this is dealt with (methodologically), including sample selection, but I think this remains an issue now for the editors to consider. The responses from the author(s) are considered and reflective and, while ultimately not leading to much change, do lead to additional qualifications. I can also accept that different fields draw on different sets of literature, conceptualizations and ultimately methods.	We appreciate the methodological differences that the reviewer has raised and appreciate as well the reflective stance that the reviewer has taken. We have developed this paper leveraging several disciplinary traditions and strengths, including urban planning, urban ecology, and urban geography, so it does not easily fall into one category. Our hope in doing so, fundamentally, was to develop an analytic frame that speaks across disciplinary boundaries and robustly reflects the reality on-the-ground for European and North American cities today – even if interpretations of what matters most may vary. We think the reviewer highlights the main finding well and we appreciate that the reviewer took time to look at our design and analysis from many angles and ask probing questions that made us reconsider and re-examine in ways that strengthened the final manuscript. In terms of the main issue raised, we appreciate that our approach draws on methods and disciplinary norms that generate the sample selection and ordering questions that we have worked through in response to the reviewer in prior revisions. We utilized a sample selection method that is well-supported and well-established, but is less quantitatively-driven and different than some fields might prefer. We did insist on this approach as it was designed to generate the most robust and meaningful outcomes we thought possible in the context of a multi-disciplinary paper – an approach we believe is important in order to understand the complexity and nuance of urban socio-environmental change and urban inequalities. Certainly, the reviewer did make us think more about the limitations of our approach and be more explicit about where qualifications were needed. We appreciate this pushback, which, we would argue, reflects the best of interdisciplinary dialogue. We consider that these qualifications and edits made along the way, in addition to responses to editors, amount to a response to the reviewer.